# On Accelerating Diffusion-Based Sampling Processes via Improved Integration Approximation

**Guoqiang Zhang**
Dept. of Computer Science
University of Exeter
United Kingdom
g.z.zhang@exeter.ac.uk

**Kenta Niwa**
Communication Science
Labs, NTT
Japan
kenta.niwa@ntt.com

**W. Bastiaan Kleijn**
School of ECS
Victoria Univ. of Wellington
New Zealand
bastiaan.kleijn@vuw.ac.nz

## Abstract

A popular approach to sample a diffusion-based generative model is to solve an ordinary differential equation (ODE). In existing samplers, the coefficients of the ODE solvers are pre-determined by the ODE formulation, the reverse discrete timesteps, and the employed ODE methods. In this paper, we consider accelerating several popular ODE-based sampling processes (including DDIM, DPM-Solver++, and EDM) by optimizing certain coefficients via improved integration approximation (IIA). We propose to minimize, for each time step, a mean squared error (MSE) function with respect to the selected coefficients. The MSE is constructed by applying the original ODE solver for a set of fine-grained timesteps, which in principle provides a more accurate integration approximation in predicting the next diffusion state. The proposed IIA technique does not require any change of a pre-trained model, and only introduces a very small computational overhead for solving a number of quadratic optimization problems. Extensive experiments show that considerably better FID scores can be achieved by using IIA-DDIM, IIA-DPM-Solver++, and IIA-EDM than the original counterparts when the neural function evaluation (NFE) is small (i.e., less than 25).

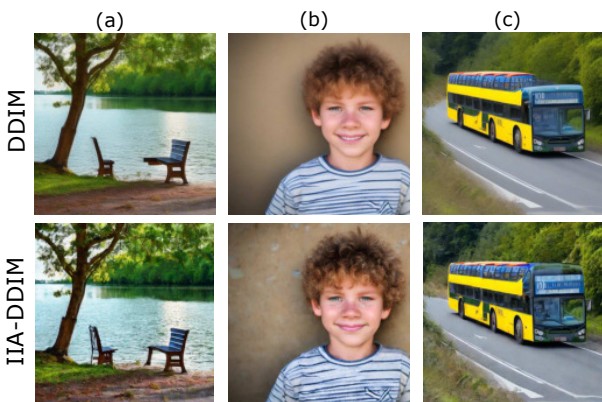

Figure 1: Comparison of DDIM and proposed IIA-DDIM with 10 timesteps for text-to-image generation over StableDiffusion V2. See Table 12 for input texts, Table 1 for FID evaluation, and Figs. 8, 9, 10 for more images.

## 1 Introduction

As one type of generative models Goodfellow et al. (2014); Arjovsky et al. (2017); Gulrajani et al. (2017); Sauer et al. (2022); Bishop (2006), diffusion probabilistic models (DPMs) have made significant progress in recent years. Following the pioneering work of Sohl-Dickstein et al. (2015), various learning and/or sampling strategies have been proposed to improve the performance of DPMs, which include, for example, denoising diffusion probabilistic models (DDPMs) Ho et al. (2020), denoising diffusion implicit models (DDIMs) Song et al. (2021a), improved DDIMs Nichol & Dhariwal (2021); Dhariwal & Nichol (2021), generalized DDIM Zhang et al. (2023b), latent diffusion models (LDMs) Rombach et al. (2022), score matching with Langevin dynamics (SMLD) Song & Ermon (2019); Song et al. (2021b;c), analytic-DPMs Bao et al. (2022b;a), optimized denoising

schedules Kingma et al. (2021); Chen et al. (2020); Lam et al. (2022), and guided diffusion strategies Nichol et al. (2022); Kim et al. (2022). It is worth noting that DDIM can be interpreted as a first-order ODE solver, where its coefficients are pre-determined by the ODE formulation and the discrete reverse timesteps. See also Yang et al. (2021) for a detailed literature overview.

To further improve the sampling qualities in DPMs, one recent research trend is to exploit high-order methods for solving the ordinary differential equations (ODEs) in the sampling processes. The authors of Liu et al. (2022) proposed pseudo linear multi-step (PLMS) sampling method, of which high-order polynomials of the estimated Gaussian noises from a score network are introduced per timestep to improve the sampling quality. The work Zhang & Chen (2022) further extends Liu et al. (2022) by refining the coefficients of the high-order polynomials of the estimated Gaussian noises, and proposes the diffusion exponential integrator sampler (DEIS). Recently, the authors of Lu et al. (2022a;b) considered solving the ODEs of a diffusion model differently from Zhang & Chen (2022). In particular, a high-order Taylor expansion of the estimated Gaussian noises was employed to approximate the continuous solutions of the ODEs more accurately. The resulting sampling methods are referred to as DPM-Solver and DPM-Solver++. The work Zhang et al. (2023a) improves the sampling performance of DDPM, DDIM, second order PLMS (S-PNDM), DEIS, and DPM-Solver by performing additional extrapolation on the estimated clean data at each reverse timestep. The recent work Karras et al. (2022) achieves state-of-the-art (SOTA) sampling performance on CIFAR10 and ImageNet64 by utilizing only the improved Euler method Ascher & Petzold (1998) to solve an ODE of a refined diffusion model, referred to as the EDM sampling procedure. Similarly to DDIM, the coefficients of EDM are pre-determined by the ODE formulation and the reverse timesteps.

In this paper, we make two main contributions. Firstly, we propose to optimize the stepsizes (or coefficients) in front of the selected gradient vectors in a number of promising ODE-based sampling processes (including DDIM, DPM-Solver++, and EDM). Our basic idea is to improve the accuracy of the integration approximation per timeslot when predicting the next diffusion state by minimising a mean squared error (MSE) function, referred to as the *improved integration approximation* (IIA) technique. The MSE per reverse timeslot is constructed by measuring the difference between the coarse- and fine-grained approximations of an ODE integration, where the fine-grained approximation is obtained by applying the original ODE solver over a set of fine-grained timesteps. The MSE is then minimized with respect to the considered stepsizes embedded in the coarse integration approximation. Our IIA technique renders more flexibility than the aforementioned existing ODE solvers, of which the update formats are always fixed for different pre-trained diffusion models. See Bar-Sinai et al. (2019) and Li et al. (2023) for related works on optimizing numerical integration parameters in the context of solving differential equations describing physical systems.

Secondly, we verify the effectiveness of IIA-DDIM, IIA-DPM-Solver++, and IIA-EDM via extensive experiments. For each method being applied to a pre-trained model with pre-defined timesteps, the optimal stepsizes are computed only once by minimising the constructed MSEs (MMSEs), and then stored for extensive sampling in the FID evaluation. To reduce computational overhead, the MMSEs are performed by solving a set of quadratic functions based on a finite number of initial noise samples. In all our experiments, introducing the IIA technique into DDIM, and DPM-Solver++, and IIA-EDM significantly improves the sampling quality for small NFEs (see Figs. 1, 3, 5, 8, 9, 10, and Table 1). Computational overhead and sampling time (see Tables 10 and 9) were measured for IIA-EDM, showing that the overhead is negligible and the sampling time is roughly the same as that of EDM.

## 2 PRELIMINARY

**Forward and reverse diffusion processes:** Suppose the data sample $\boldsymbol{x} \in \mathbb{R}^d$ follows a data distribution $p_{data}(\boldsymbol{x})$ with a bounded variance. A forward diffusion process progressively adds Gaussian noises to the data samples $\boldsymbol{x}$ to obtain $\boldsymbol{z}_t$ as $t$ increases from 0 until $T$. The conditional distribution of $\boldsymbol{z}_t$ given $\boldsymbol{x}$ can be represented as

$$q_{t|0}(\boldsymbol{z}_t|\boldsymbol{x}) = \mathcal{N}(\boldsymbol{z}_t|\alpha_t\boldsymbol{x}, \sigma_t^2\boldsymbol{I}), \tag{1}$$

where $\alpha_t$ and $\sigma_t$ are assumed to be differentiable functions of $t$ with bounded derivatives. We use $q(\boldsymbol{z}_t; \alpha_t, \sigma_t)$ to denote the marginal distribution of $\boldsymbol{z}_t$. The samples of the distribution $q(\boldsymbol{z}_T; \alpha_T, \sigma_T)$ would be practically indistinguishable from pure Gaussian noises if $\sigma_T \gg \alpha_T$.

The reverse process of a diffusion model firstly draws a sample $\boldsymbol{z}_T$ from $\mathcal{N}(\boldsymbol{0}, \sigma_T^2\boldsymbol{I})$, and then progressively denoises it to obtain a sequence of diffusion states $\{\boldsymbol{z}_{t_i} \sim p(\boldsymbol{z}; \alpha_{t_i}, \sigma_{t_i})\}_{i=0}^N$, where

we use the notation $p(\cdot)$ to indicate that reverse sample distribution might not be identical to the forward distribution $q(\cdot)$ because of practical approximations. It is expected that the final sample $z_{t_N}$ is roughly distributed according to $p_{data}(\boldsymbol{x})$, i.e., $p_{data}(\boldsymbol{x}) \approx p(\boldsymbol{z}_{t_N}; \alpha_{t_N}, \sigma_{t_N})$ where $t_N = 0$.

**ODE formulation:** In Song et al. (2021c), Song et al. present a so-called *probability flow* ODE which shares the same marginal distributions as $\boldsymbol{z}_t$ in (1). Specifically, with the formulation (1) for a forward diffusion process, its reverse ODE form can be represented as

$$d\boldsymbol{z} = \underbrace{\left[ \frac{d\log\alpha_t}{dt}\boldsymbol{z}_t - \frac{1}{2}\left[\frac{d\sigma_t^2}{dt} - 2\frac{d\log\alpha_t}{dt}\sigma_t^2\right]\nabla_{\boldsymbol{z}}\log q(\boldsymbol{z}_t; \alpha_t, \sigma_t)\right]}_{\boldsymbol{d}(\boldsymbol{z}, t)} dt, \quad (2)$$

where $\nabla_{\boldsymbol{z}}\log q(\boldsymbol{z}; \alpha_t, \sigma_t)$ in (2) is the score function Hyvarinen (2005) pointing towards higher density of data samples at the given noise level $(\alpha_t, \sigma_t)$, and the gradient $\frac{d\boldsymbol{z}}{dt}$ is represented by $\boldsymbol{d}(\boldsymbol{z}, t)$.

As $t$ increases, the probability flow ODE (2) continuously reduces noise level of the data samples in the reverse process. In the ideal scenario where no approximations are introduced in (2), the sample distribution $p(\boldsymbol{z}; \alpha_t, \sigma_t)$ approaches $p_{data}(\boldsymbol{x})$ as $t$ goes from $T$ to 0. As a result, the sampling process of a diffusion model boils down to solving the ODE form (2), where randomness is only introduced in the initial samples. This has opened up the research opportunity of exploiting different ODE solvers in diffusion-based sampling processes.

**Denoising score matching:** To be able to utilize (2) for sampling, one needs to specify a particular form of the score function $\nabla_{\boldsymbol{z}}\log q(\boldsymbol{z}; \alpha_t, \sigma_t)$. One common approach is to train a noise estimator $\hat{\boldsymbol{\epsilon}}_{\boldsymbol{\theta}}$ by minimizing the expected $L_2$ error for samples drawn from $q_{data}$ (see Ho et al. (2020); Song et al. (2021c;a)):

$$\mathbb{E}_{\boldsymbol{x}\sim p_{data}}\mathbb{E}_{\boldsymbol{\epsilon}\sim\mathcal{N}(\boldsymbol{0}, \sigma_t^2\boldsymbol{I})}\|\hat{\boldsymbol{\epsilon}}_{\boldsymbol{\theta}}(\alpha_t\boldsymbol{x} + \sigma_t\boldsymbol{\epsilon}, t) - \boldsymbol{\epsilon}\|_2^2, \quad (3)$$

where $(\alpha_t, \sigma_t)$ are from the forward process (1). With (3), the score function can then be represented in terms of $\hat{\boldsymbol{\epsilon}}_{\boldsymbol{\theta}}(\boldsymbol{z}_t; t)$ as

$$\nabla_{\boldsymbol{z}}\log q(\boldsymbol{z}_t; \alpha_t, \sigma_t) = -\hat{\boldsymbol{\epsilon}}_{\boldsymbol{\theta}}(\boldsymbol{z}_t; t)/\sigma_t. \quad (4)$$

# 3 IMPROVED INTEGRATION APPROXIMATION (IIA) FOR DDIM AND DPM-SOLVER++

In this section, we first briefly present the basic principle for incorporating IIA into existing ODE solvers. After that, we consider applying the IIA technique for both the conventional DDIM sampling and the classifier-free guided DDIM sampling developed for text-to-image generation. In the end, we explain how to design IIA-DPM-Solver++ for text-to-image generation.

**Design principles for incorporating IIA into an ODE solver:** To our best knowledge, the formats of existing diffusion samplers (e.g., DDIM, EDM and DPM-Solver++) are fixed irrespective of the number of time steps and particular pre-trained models. It is likely that the coefficients of those ODE solvers are not always optimal. To improve the accuracy of the integration approximation per timestep in an existing ODE solver, we propose to optimize the stepsizes (or coefficients) in front of certain selected quantities via MMSE. The quantities can be functions of, for example, estimated Gaussian noises and/or estimated clean data. The MSE per time-interval can be constructed by measuring the difference between the coarse- and fine-grained integration approximations, where the stepsizes to be optimized are embedded in the coarse-grained integration approximation. In principle, the selected quantities are appropriate as long as the residual error of the MSE after stepsize optimization is reduced notably compared to that of the original ODE solver. As will be presented in the following, when we incorporate IIA into DDIM, DPM-Solver++, and EDM by following the above guidance, we focus on a particular proper selection of the quantities for each sampling method instead of finding the optimal configuration of these quantities that produces the smallest residual error.

**IIA for conventional DDIM sampling :** The conventional DDIM sampling procedure is in fact a first-order solver for the ODE formulation (2) (see Lu et al. (2022a); Zhang & Chen (2022)). Its update expression is given by

$$\boldsymbol{z}_{i+1} = \alpha_{t_{i+1}}\overbrace{\left(\frac{\boldsymbol{z}_i - \sigma_{t_i}\hat{\boldsymbol{\epsilon}}_{\boldsymbol{\theta}}(\boldsymbol{z}_i, t_i)}{\alpha_{t_i}}\right)}^{\hat{\boldsymbol{x}}(\boldsymbol{z}_i, t_i)} + \sigma_{t_{i+1}}\hat{\boldsymbol{\epsilon}}_{\boldsymbol{\theta}}(\boldsymbol{z}_i, t_i) \approx \boldsymbol{z}_i + \int_{t_i}^{t_{i+1}}\boldsymbol{d}(\boldsymbol{z}, \tau)d\tau, \quad (5)$$

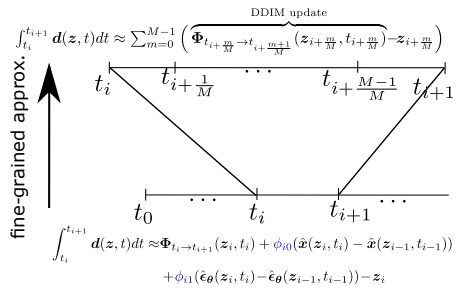

Figure 2: Coarse and fine-grained approximations of the integration $\int_{t_i}^{t_{i+1}} \boldsymbol{d}(\boldsymbol{z},t)dt$. $\{\phi_{i0}, \phi_{i1}\}$ are the introduced stepsizes in IIA-DDIM for conventional sampling, which are determined by (7).

where $\hat{\boldsymbol{x}}(\boldsymbol{z}_i, t_i)$ denotes an estimator for the clean image $\boldsymbol{x}$, and $\boldsymbol{d}(\boldsymbol{z}, \tau)$ is from (2).

We now introduce two additional quantities in computing $\boldsymbol{z}_{i+1}$, which are given by

$$
\boldsymbol{z}_{i+1} = \overbrace{\alpha_{t_{i+1}}\hat{\boldsymbol{x}}(\boldsymbol{z}_i, t_i) + \sigma_{t_{i+1}}\hat{\boldsymbol{\epsilon}}_{\boldsymbol{\theta}}(\boldsymbol{z}_i, t_i)}^{\boldsymbol{\Phi}_{t_i \to t_{i+1}}(\boldsymbol{z}_i, t_i)} + \phi_{i0}^*\overbrace{(\hat{\boldsymbol{x}}(\boldsymbol{z}_i, t_i) - \hat{\boldsymbol{x}}(\boldsymbol{z}_{i-1}, t_{i-1}))}^{\text{1st quantity}}
$$
$$
+ \phi_{i1}^*\underbrace{(\hat{\boldsymbol{\epsilon}}_{\boldsymbol{\theta}}(\boldsymbol{z}_i, t_i) - \hat{\boldsymbol{\epsilon}}_{\boldsymbol{\theta}}(\boldsymbol{z}_{i-1}, t_{i-1}))}_{\text{2nd quantity}}. \tag{6}
$$

The first quantity in (6) can be interpreted as a gradient vector pointing towards the data sample $\boldsymbol{x}$ (see Zhang et al. (2023a)). The second quantity in (6) is a vector measuring the difference of the noise estimators at timesteps $t_i$ and $t_{i-1}$, which is inspired by high-order ODE solvers Zhang & Chen (2022); Liu et al. (2022). The two stepsizes $(\phi_{i0}^*, \phi_{i1}^*)$ in front of the two quantities in (6) are computed by the MMSE as follows:

$$
(\phi_{i0}^*, \phi_{i1}^*) = \arg\min_{\phi_{i0}, \phi_{i1}} \mathbb{E}_{\boldsymbol{z}_{t_0} \sim \mathcal{N}(0, \sigma_T^2 \boldsymbol{I})} \Big\| \boldsymbol{\Phi}_{t_i \to t_{i+1}}(\boldsymbol{z}_i, t_i) + \phi_{i0}(\hat{\boldsymbol{x}}(\boldsymbol{z}_i, t_i) - \hat{\boldsymbol{x}}(\boldsymbol{z}_{i-1}, t_{i-1}))
$$
$$
+ \phi_{i1}(\hat{\boldsymbol{\epsilon}}_{\boldsymbol{\theta}}(\boldsymbol{z}_i, t_i) - \hat{\boldsymbol{\epsilon}}_{\boldsymbol{\theta}}(\boldsymbol{z}_{i-1}, t_{i-1})) - \boldsymbol{z}_i - \sum_{m=0}^{M-1}\Big(\boldsymbol{\Phi}_{t_{i+\frac{m}{M}} \to t_{i+\frac{m+1}{M}}}(\boldsymbol{z}_{i+\frac{m}{M}}, t_{i+\frac{m}{M}}) - \boldsymbol{z}_{i+\frac{m}{M}}\Big) \Big\|^2, \tag{7}
$$

where the expectation is over the probability distribution of the initial state $\boldsymbol{z}_{t_0}$. The summation in the RHS of (7) from $m=0$ until $m=M-1$ corresponds to applying DDIM over a fine-grained set of timesteps $\{t_{i+\frac{m}{M}}\}_{m=0}^m$ within the time-interval of $[t_i, t_{i+1}]$ (see Fig. 2). In principle, when $M$ goes to infinity, the summation would provide a very accurate approximation of the integration in (7). The solution $(\phi_{i0}^*, \phi_{i1}^*)$ makes the update $\boldsymbol{z}_{i+1}$ in (6) optimal with respect to the MMSE criterion of (7) over the probability distribution of the initial state $\boldsymbol{z}_{t_0}$. We note that the FID score for measuring image quality is in fact also performed over the probability distribution of the initial state. In principle, if the MSE on the RHS of (7) is indeed reduced due to $\{\phi_{i0}^*, \phi_{i1}^*\}$, the resulting FID would be improved. In practice, the expectation in (7) can be realized by utilizing a set $\mathcal{B}$ of initial samples at $t_0$. $(\phi_{i0}^*, \phi_{i1}^*)$ can then be computed by solving a quadratic optimization problem. Once the optimal stepsizes are obtained, they can be stored and re-used later on for extensive sampling.

**IIA for classifier-free guided DDIM sampling for text-to-image generation:** The classifier-free guided DDIM method has been widely used in diffusion based text-to-image generation. The basic idea is to evaluate the noise prediction model two times at each timestep $t_i$, the first time with a text prompt: $\hat{\boldsymbol{\epsilon}}_{\boldsymbol{\theta}}(\boldsymbol{z}_i, \phi = P; t_i)$ (where $P$ denotes the text prompt) and the second time with the null text prompt: $\hat{\boldsymbol{\epsilon}}_{\boldsymbol{\theta}}(\boldsymbol{z}_i, \phi = null; t_i)$. The two predicted noises are then combined to obtain a refined noise $\ddot{\boldsymbol{\epsilon}}_{\boldsymbol{\theta}}(\boldsymbol{z}_i, \phi = P; t_i)$, which is plugged into the DDIM update in computing the next diffusion state.

To apply IIA to the above text-to-image generation scenario, we optimize the stepsize (or coefficient) in front of $\ddot{\boldsymbol{\epsilon}}_{\boldsymbol{\theta}}(\boldsymbol{z}_i, \phi = P; t_i)$, which is found to be preferable over the two terms in (6) for the conventional DDIM method:

$$
\boldsymbol{z}_{i+1} = \overbrace{\boldsymbol{\Phi}_{t_i \to t_{i+1}}(\boldsymbol{z}_i, t_i)}^{\text{DDIM}} + \beta_i \ddot{\boldsymbol{\epsilon}}_{\boldsymbol{\theta}}(\boldsymbol{z}_i, \phi = P; t_i), \tag{8}
$$

where $\beta_i$ is the introduced stepsize and $\ddot{\boldsymbol{\epsilon}}_{\boldsymbol{\theta}}(\boldsymbol{z}_i, \phi = P; t_i)$ is utilized in the DDIM update expression.

To optimize the stepsize $\beta_i$ in (8), we construct an MSE estimate that averages over the probability distributions of both the initial noise vector $\boldsymbol{z}_{t_0}$ and the text prompts. We assume that the text prompts

follow a non-parametric distribution that can be approximated by sampling. The MSE can then be approximated as a quadratic function of $\beta_i$ by using finite samples of text prompts and $\boldsymbol{z}_{t_0}$.

**IIA for classifier-free guided DPM-Solver++ sampling for text-to-image generation:** IIA-DPM-Solver++ can be designed in a similar way as IIA-DDIM for text-to-image generation presented above. Again the MSE for computing the optimal stepsizes in IIA-DPM-Solver++ should take into account the probability distributions of $\boldsymbol{z}_{t_0}$ and the text prompts. See Appendix C for details.

**Remark 1.** *We have also considered the application of IIA to the high-order methods SPNDM and IPNDN Zhang & Chen (2022). See the performance results in Appendix G. In summary, the IIA technique improves the sampling performance of SPNDM and IPNDM for certain pre-trained models.*

## 4 IMPROVED INTEGRATION APPROXIMATION (IIA) FOR EDM

In this section, we first briefly review the EDM sampling procedure for solving the ODE (2) in Karras et al. (2022), which produces SOTA performance over CIFAR10 and ImageNet64. We then present our new IIA technique for solving the ODE more accurately, thus accelerating the sampling process.

### 4.1 REVIEW OF EDM SAMPLING PROCEDURE

The recent work Karras et al. (2022) reparameterizes the forward diffusion process (1) to be

$$q_{t|0(\boldsymbol{z}_t|\boldsymbol{x})} = \mathcal{N}(\boldsymbol{z}_t|\alpha_t\boldsymbol{x}, \alpha_t^2\tilde{\sigma}_t^2\boldsymbol{I}), \qquad (9)$$

where $\sigma_t$ of (1) is represented as $\sigma_t = \alpha_t\tilde{\sigma}_t$. Let $\boldsymbol{D}_{\boldsymbol{\theta}}(\boldsymbol{z}_t, t)$ denote an estimator for the data sample $\boldsymbol{x}$ at timestep $t$. It can be computed in terms of the noise estimator $\hat{\boldsymbol{\epsilon}}_{\boldsymbol{\theta}}$ as $\boldsymbol{D}_{\boldsymbol{\theta}}(\boldsymbol{z}_t, t) = \boldsymbol{z}_t/\alpha_t - \tilde{\sigma}_t\hat{\boldsymbol{\epsilon}}_{\boldsymbol{\theta}}(\boldsymbol{z}_t, t)$. The resulting probability flow ODE takes the form of

$$d\boldsymbol{z} = \underbrace{\left[\left(\frac{\dot{\alpha}_t}{\alpha_t} + \frac{\dot{\tilde{\sigma}}_t}{\tilde{\sigma}_t}\right)\boldsymbol{z} - \frac{\dot{\tilde{\sigma}}_t\alpha_t}{\tilde{\sigma}_t}\boldsymbol{D}_{\boldsymbol{\theta}}(\boldsymbol{z}_t, t)\right]}_{\boldsymbol{d}(\boldsymbol{z},t)}dt, \qquad (10)$$

where the dot operation denotes a time derivative.

The work Karras et al. (2022) proposed a deterministic sampling procedure for solving (10) for arbitrary $\tilde{\sigma}_t$ and $\alpha_t$. Basically, the improved Euler method Ascher & Petzold (1998) was utilized for solving the ODE form. The resulting update expressions from time $t_i$ to $t_{i+1}$ are given by

$$\tilde{\boldsymbol{z}}_{i+1} = \boldsymbol{z}_i + (t_{i+1} - t_i)\boldsymbol{d}_i, \qquad (11)$$

$$\boldsymbol{z}_{i+1} = \boldsymbol{z}_i + \underbrace{(t_{i+1} - t_i)(\frac{1}{2}\boldsymbol{d}_i + \frac{1}{2}\boldsymbol{d}'_{i+1|i})}_{\approx \int_{t_i}^{t_{i+1}}\boldsymbol{d}(\boldsymbol{z},t)dt}, \qquad (12)$$

where $(t_{i+1} - t_i)$ is the stepsize, $\boldsymbol{d}_i = \boldsymbol{d}(\boldsymbol{z}_i, t_i)$, and $\boldsymbol{d}'_{i+1|i} = \boldsymbol{d}(\tilde{\boldsymbol{z}}_{i+1}, t_{i+1})$. $\tilde{\boldsymbol{z}}_{i+1}$ is the intermediate estimate of the hidden state $\boldsymbol{z}$ at time $t_{i+1}$. The final estimate $\boldsymbol{z}_{i+1}$ is computed by utilizing the average of the gradients $\boldsymbol{d}_i$ and $\boldsymbol{d}'_{i+1|i}$. We will explain in the next subsection how we compute the optimal stepsize in (12) instead of using the fixed one $(t_{i+1} - t_i)$.

### 4.2 BASIC IIA FOR EDM (BIIA-EDM) VIA MMSE

In this subsection, we consider improving the accuracy of the integral approximation in (12) at timestep $t_i$. To do so, we propose to approximate the integration $\int_{t_i}^{t_{i+1}}\boldsymbol{d}(\boldsymbol{z}, t)dt$ by utilizing the most recent set of gradients $\{\frac{1}{2}\boldsymbol{d}_{i-k} + \frac{1}{2}\boldsymbol{d}'_{i-k+1|i-k}\}_{k=0}^r$, given by

$$\int_{t_i}^{t_{i+1}}\boldsymbol{d}(\boldsymbol{z}, t)dt \approx \sum_{k=0}^r \gamma_{ik}\underbrace{(\frac{1}{2}\boldsymbol{d}_{i-k} + \frac{1}{2}\boldsymbol{d}'_{i-k+1|i-k})}_{\boldsymbol{\Delta}_i(\boldsymbol{z}_{i-k})}, \qquad (13)$$

where the set of coefficients $\{\gamma_{ik}\}_{k=0}^r$ can be interpreted as the stepsizes being multiplied by those gradients. We attempt to find a proper choice for $\{\gamma_{ik}\}_{k=0}^r$ so that the integral approximation (13) will become more accurate than the one in (12).

Our motivation for utilizing the most recent $r+1$ gradients in (13) is inspired by SGD with momentum Sutskever et al. (2013); Polyak (1964) and its variants Kingma & Ba (2017); Zhang (2023) which

computes and makes use of the exponential moving average of historical gradients in updating machine learning models. In general, the recent gradients provide additional directions pointing towards higher functional values. Proper exploration of those gradients can help to accelerate the diffusion sampling process.

We are now in a position to compute $\{\gamma_{ik}\}_{k=0}^{r}$ in (13) at timestep $t_i$. Our basic idea is to first obtain a highly accurate approximation of $\int_{t_i}^{t_{i+1}} \boldsymbol{d}(\boldsymbol{z}, t)dt$, and then compute proper values of $\{\gamma_{ik}\}_{k=0}^{r}$ so that the coarse approximation (13) is optimally close to the accurate approximation. Similarly to the design of IIA-DDIM, we approximate the integration $\int_{t_i}^{t_{i+1}} \boldsymbol{d}(\boldsymbol{z}, t)dt$ by applying the improved Euler method over a set of fine-grained timesteps $\{t_{i+\frac{m}{M}}\}_{m=0}^{M}$, where $m = 0$ and $m = M$ correspond to the starting time $t_i$ and ending time $t_{i+1}$, respectively. Mathematically, the integration $\int_{t_i}^{t_{i+1}} \boldsymbol{d}(\boldsymbol{z}, t)dt$ can be estimated more accurately over $\{t_{i+\frac{m}{M}}\}_{m=0}^{M}$ as

$$\int_{t_i}^{t_{i+1}} \boldsymbol{d}(\boldsymbol{z}, t)dt = \sum_{m=0}^{M-1} \int_{t_{i+\frac{m}{M}}}^{t_{i+\frac{m+1}{M}}} \boldsymbol{d}(\boldsymbol{z}, t)dt \approx \sum_{m=0}^{M-1} \left( t_{i+\frac{m+1}{M}} - t_{i+\frac{m}{M}} \right) \left( \frac{1}{2}\boldsymbol{d}_{i+\frac{m}{M}} + \frac{1}{2}\boldsymbol{d}'_{i+\frac{m+1}{M}|i+\frac{m}{M}} \right)$$
$$= \boldsymbol{\Delta}_{fg}(\boldsymbol{z}_i), \tag{14}$$

where we use $\boldsymbol{\Delta}_{fg}(\boldsymbol{z}_i)$ to denote the summation of the fine-grained integration approximations.

We compute the optimal solution of $\{\gamma_{ik}\}_{k=0}^{r}$ in (13) via MMSE with regard to the difference of the two approximations $\sum_{k=0}^{r} \gamma_{ik}\boldsymbol{\Delta}_i(\boldsymbol{z}_{i-k})$ and $\boldsymbol{\Delta}_{fg}(\boldsymbol{z}_i)$:

$$\{\gamma_{ik}^*\}_{k=0}^{r} = \arg\min \mathbb{E}_{\boldsymbol{z}_{t_0} \sim \mathcal{N}(0, \sigma_T^2 \boldsymbol{I})} \left\| \sum_{k=0}^{r} \gamma_{ik}\boldsymbol{\Delta}_i(\boldsymbol{z}_{i-k}) - \boldsymbol{\Delta}_{fg}(\boldsymbol{z}_i) \right\|^2, \tag{15}$$

where $\{\boldsymbol{z}_{i-k}\}_{k=0}^{r}$ are determined by the initial state $\boldsymbol{z}_{t_0}$ in the deterministic sampling procedure.

Similarly to the stepsize optimization in IIA-DDIM, one can solve the optimization problem (15) by utilizing a set $\mathcal{B}$ of initial samples at timestep $t_0$ to approximate the expectation operation. The solution $\{\gamma_{ik}^*\}_{k=0}^{r}$ can then be easily computed by minimizing a quadratic function. Consider the simple case of $r = 0$ as an example, where only the quantity $\boldsymbol{\Delta}_i(\boldsymbol{z}_i)$ is employed in (15). The optimal solution $\gamma_{i0}^*$ is easily seen to be

$$\gamma_{i0}^* \approx \frac{\sum_{\boldsymbol{z}_{t_0} \in \mathcal{B}} \langle \boldsymbol{\Delta}_i(\boldsymbol{z}_i), \boldsymbol{\Delta}_{fg}(\boldsymbol{z}_i) \rangle}{\sum_{\boldsymbol{z}_{t_0} \in \mathcal{B}} \|\boldsymbol{\Delta}_i(\boldsymbol{z}_i)\|^2}, \tag{16}$$

where $\langle \cdot \rangle$ denotes inner product. For the general case of $r > 0$, one can also easily derive the closed-form solution for $\{\gamma_{ik}^*\}_{k=0}^{r}$. see Alg. 2 in appendix for the sampling procedure of BIIA-EDM after stepsize optimization.

## 4.3 ADVANCED IIA FOR EDM VIA MMSE

In this subsection, we present an advanced IIA technique for EDM. To do so, we reformulate the update expression for $\boldsymbol{z}_{i+1}$ in (12). The resulting update expression is summarized in a lemma below:

**Lemma 1.** *The update expression for $\boldsymbol{z}_{i+1}$ at timestep $t_i$ in EDM under the configuration of $\alpha_t = 1$ can be reformulated to be*

$$\boldsymbol{z}_{i+1} = \boldsymbol{z}_i + \underbrace{\frac{t_{i+1} - t_i}{t_i}}_{\text{1st stepsize}} \underbrace{[\boldsymbol{z}_i - \boldsymbol{D}_{\boldsymbol{\theta}}(\boldsymbol{z}_i, t_i)]}_{\text{1st quantity}} + \underbrace{\frac{(t_{i+1} - t_i)}{2t_{i+1}}}_{\text{2nd stepsize}} \underbrace{[\boldsymbol{D}_{\boldsymbol{\theta}}(\boldsymbol{z}_i; t_i) - \boldsymbol{D}_{\boldsymbol{\theta}}(\tilde{\boldsymbol{z}}_{i+1}; t_{i+1})]}_{\text{2nd quantity}}, \tag{17}$$

*where the detailed derivation is provided in Appendix B.*

Lemma 1 indicates that the integration approximation over the time interval $[t_i, t_{i+1}]$ for computing $\boldsymbol{z}_{i+1}$ is realized as a weighted summation of two quantities (see Zhang et al. (2023a)): $[\boldsymbol{z}_i - \boldsymbol{D}_{\boldsymbol{\theta}}(\boldsymbol{z}_i, t_i)]$ and $[\boldsymbol{D}_{\boldsymbol{\theta}}(\tilde{\boldsymbol{z}}_{i+1}; t_{i+1}) - \boldsymbol{D}_{\boldsymbol{\theta}}(\boldsymbol{z}_i; t_i)]$. The two stepsizes in front of the two quantities in (17) are functions of $t_i$ and $t_{i+1}$, which are predetermined by the improved Euler method.

We propose to introduce new stepsizes in front of the two quantities in (17). As will be explained below, the new stepsizes will be determined by the improved integration approximation (IIA) technique.

---

**Algorithm 1** IIA-EDM as an extension of EDM in Karras et al. (2022)

1: **Input:**
2:   number of time steps $N, r = 1, \{(\beta_{ik}^{\boldsymbol{\epsilon},*}, \beta_{ik}^{\boldsymbol{D},*})|k = 0, \ldots, r\}_{i=1}^{N-2}$  [ pre-computed values via MMSE]
3: **Sample** $\boldsymbol{z}_0 \sim \mathcal{N}(\boldsymbol{0}, \alpha_{t_0}^2 \tilde{\sigma}_{t_0}^2 \boldsymbol{I})$
4: **for** $i \in \{0, 1, \ldots, N-1\}$ **do**
5:     $\boldsymbol{d}_i \leftarrow \boldsymbol{d}_i(\boldsymbol{z}_i, t_i)$
6:     $\tilde{\boldsymbol{z}}_{i+1} \leftarrow \boldsymbol{z}_i + (t_{i+1} - t_i)\boldsymbol{d}_i$
7:     **if** $\sigma_{t_{i+1}} \neq 0$ **then**
8:         $\boldsymbol{d}'_{i+1|i} \leftarrow \boldsymbol{d}(\tilde{\boldsymbol{z}}_{i+1}, t_{i+1})$
9:         $\boldsymbol{z}_{i+1} \leftarrow \boldsymbol{z}_i + \sum_{k=0}^{r} \left[ \beta_{ik}^{\boldsymbol{\epsilon},*} [\boldsymbol{z}_{i-k} - \boldsymbol{D}_{\boldsymbol{\theta}}(\boldsymbol{z}_{i-k}; t_{i-k})] + \beta_{ik}^{\boldsymbol{D},*} \left[ \boldsymbol{D}_{\boldsymbol{\theta}}(\boldsymbol{z}_{i-k}; t_{i-k}) - \boldsymbol{D}_{\boldsymbol{\theta}}(\tilde{\boldsymbol{z}}_{i-k+1}; t_{i-k+1}) \right] \right]$
10:     **end if**
11: **end for**
12: **Output:** $\boldsymbol{z}_N$

---

Specifically, we approximate the integration $\int_{t_i}^{t_{i+1}} \boldsymbol{d}(\boldsymbol{z}, t)dt$ at timestep $t_i$ as

$$\int_{t_i}^{t_{i+1}} \boldsymbol{d}(\boldsymbol{z}, t)dt \approx \sum_{k=0}^{r} \left[ \beta_{ik}^{\boldsymbol{\epsilon}}[\boldsymbol{z}_{i-k} - \boldsymbol{D}_{\boldsymbol{\theta}}(\boldsymbol{z}_{i-k}; t_{i-k})] + \beta_{ik}^{\boldsymbol{D}} \left[ \boldsymbol{D}_{\boldsymbol{\theta}}(\boldsymbol{z}_{i-k}; t_{i-k}) - \boldsymbol{D}_{\boldsymbol{\theta}}(\tilde{\boldsymbol{z}}_{i-k+1}; t_{i-k+1}) \right] \right]$$
$$= S_i \left( \{\beta_{ik}^{\boldsymbol{\epsilon}}, \beta_{ik}^{\boldsymbol{D}}\}_{k=0}^{r} \right). \tag{18}$$

It is noted that a number of the most recent quantities are also included in (18) for the purpose of providing additional gradient directions in the MSE minimisation.

Next we compute the optimal stepsizes in the above function $S_i(\cdot)$ by the following MMSE:

$$\{\beta_{i0}^{\boldsymbol{\epsilon},*}, \beta_{i1}^{\boldsymbol{D},*}\}_{k=0}^{r} = \arg\min \mathbb{E}_{\boldsymbol{z}_{t_0} \sim \mathcal{N}(0, \sigma_T^2 \boldsymbol{I})} \|S_i \left( \{\beta_{ik}^{\boldsymbol{\epsilon}}, \beta_{ik}^{\boldsymbol{D}}\}_{k=0}^{r} \right) - \boldsymbol{\Delta}_{fg}(\boldsymbol{z}_i)\|^2, \tag{19}$$

where $\boldsymbol{\Delta}_{fg}(\boldsymbol{z}_i)$ is from (14). Since $S_i(\cdot)$ is a linear function of its variables, the optimal solution in (19) can be easily computed by minimizing a quadratic function. Similarly to the earlier subsection, the expectation operation in (19) can be approximated by utilizing a set $\mathcal{B}$ of initial samples at timestep $t_0$. Again the optimisation (19) only needs to be performed once, and then can be used for extensive sampling.

By inspection of (15), (18) and (19), we can conclude that the optimisation (19) exploits the internal structure of the update for $\boldsymbol{z}_{i+1}$ in EDM. For the special case of $r = 0$, (19) involves two variables $(\beta_{i0}^{\boldsymbol{\epsilon}}, \beta_{i0}^{\boldsymbol{D}})$ while (15) only consists of one variable $\gamma_{i0}$. Informally, the residual error of (19) after minimisation should be smaller than that of (15), which would lead to improved sampling quality.

## 5 EXPERIMENTS

We investigated the performance gain of the IIA technique when being incorporated into DDIM, and DPM-Solver++, and EDM sampling procedures. For the EDM sampling procedure, two IIA techniques are proposed. The associated sampling procedures are referred to as BIIA-EDM (see Alg. 2) and IIA-EDM. As we mentioned earlier, the optimal stepsizes for each pre-trained model over a particular set of reverse timesteps were only computed once and were then stored and used for generating a default of 50K images (unless specified otherwise) in the computation of the FID score. It is found that the IIA technique significantly improves the sampling qualities for low NFEs (e.g., less than 25). For text-to-image generation, IIA-DDIM and IIA-DPM-Solver significantly outperforms their counterparts.

### 5.1 PERFORMANCE OF BIIA-EDM AND IIA-EDM

In this experiment, we tested four pre-trained models for four datasets: CIFAR10, FFHQ, AFHQV2, and ImageNet64 (see Table 2 in Appendix D.1). The set-size $|\mathcal{B}|$ for computing the optimal stepsizes when employing the IIA techniques was $|\mathcal{B}| = 200$, which is also the default minibatch size for sampling in the EDM official open-source repository.[1] See Table 2 for the setup of other hyperparameters. We note that $r$ in BIIA-EDM and IIA-EDM was set to $r = 1$ to save memory space.

---

[1] https://github.com/NVlabs/edm

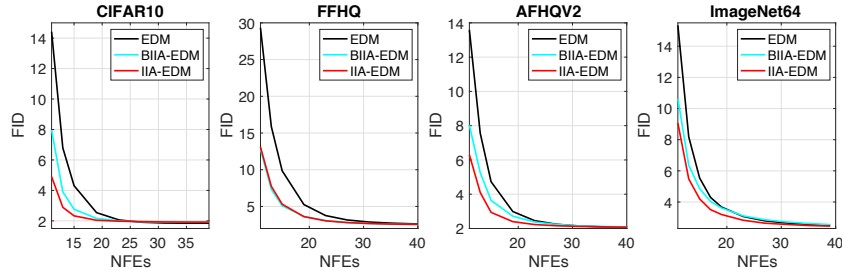

Figure 3: Sampling performance of EDM, BIIA-EDM, and IIA-EDM over four datasets.

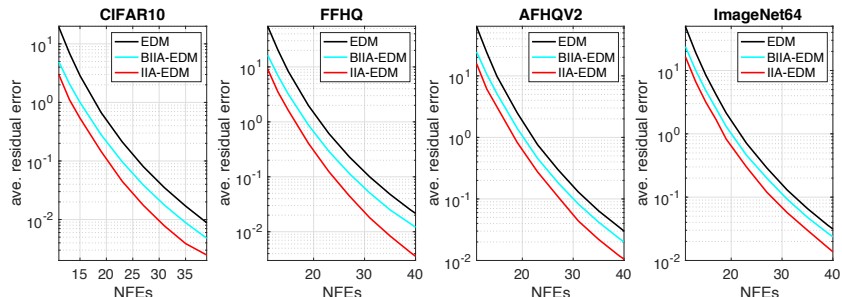

Figure 4: Comparison of average residual errors of EDM, BIIA-EDM, IIA-EDM in Fig. 3.

Table 1: Comparison of five methods for text-to-image generation over StableDiffusion V2 in terms of FID (the lower the better) and CLIP (the high the better) scores.

| | | DDIM | IIA-DDIM | DPM -Solver++ | IIA-DPM -Solver++ | PLMS | | | DDIM | IIA-DDIM | DPM -Solver++ | IIA-DPM -Solver++ | PLMS |
|---|---|---|---|---|---|---|---|---|---|---|---|---|---|
| 10 NFEs | FID | 14.78 | 13.21 | 15.82 | 12.97 | 24.42 | 30 NFEs | FID | 15.08 | 14.03 | 14.23 | 13.26 | 15.31 |
| | ClIP | 24.86 | 24.93 | 24.83 | 25.32 | 23.85 | | CLIP | 25.00 | 25.05 | 25.05 | 25.16 | 24.92 |
| 20 NFEs | FID | 14.65 | 13.14 | 13.85 | 12.77 | 14.30 | 40 NFEs | FID | 14.69 | 13.85 | 14.29 | 13.68 | 15.05 |
| | ClIP | 25.00 | 25.08 | 25.02 | 25.21 | 24.80 | | CLIP | 25.01 | 25.05 | 25.05 | 25.12 | 24.95 |

Fig. 3 visualizes the FID scores for the four pre-trained models. It is clear that IIA-EDM consistently outperforms the EDM sampling procedure when the NFE is smaller than 25. This can be explained by the fact that for a small NFE, the integration approximation in EDM is not accurate. IIA-EDM improves the accuracy of the integration approximation by introducing optimal stepsizes. The performance of IIA-EDM is also superior to that of BIIA-EDM because IIA-EDM exploits the internal structure of the EDM update expressions (see Lemma 1 and (18)), making it more flexible.

Fig. 4 displays the average residual errors between the coarse- and fine-grained integration approximations for EDM, BIIA-EDM, and IIA-EDM. It is clear that for each NFE, IIA-EDM provides the smallest error while EDM yields the largest error. This indicates that the original stepsizes of EDM are not optimal for at least small NFEs. Our work provides one approach to compute better stepsizes via IIA for small NFEs.

It is seen from the FID curves over ImageNet64 in Fig. 3 that BIIA-EDM performs slightly worse than EDM when NFE is greater than 25. This may be because, for large NFEs, an accurate integration approximation does not necessarily lead to a better sampling quality (e.g., see the sampling performance of Bao et al. (2022a)). To the best of our knowledge, it is not clear from the literature why for large NFEs, there exists a discrepancy between FID scores and accurate integration approximation.

**Sampling time and computational overhead of IIA-EDM:**    The sampling time of IIA-EDM and EDM can be found in Table 9 in Appendix E. It can be concluded from the table that the two methods consume almost the same amount of time per mini-batch, demonstrating the efficiency of IIA-EDM. The computational overhead of IIA-EDM is summarized in Table 10. It is seen the time overhead is very small in comparison to the training or fine-tuning of a typical DNN model.

### 5.2   EVALUATION OF IIA-DDIM AND IIA-DPM-SOLVER++

**Text-to-image generation:**    In this experiment, we performed FID and CLIP evaluation for IIA-DDIM, IIA-DPM-Solver++, DDIM, DPM-Solver++, and PLMS by using the validation set of COCO2014 over StableDiffusion V2. For each sampling method, 20K images of size $512 \times 512$ were

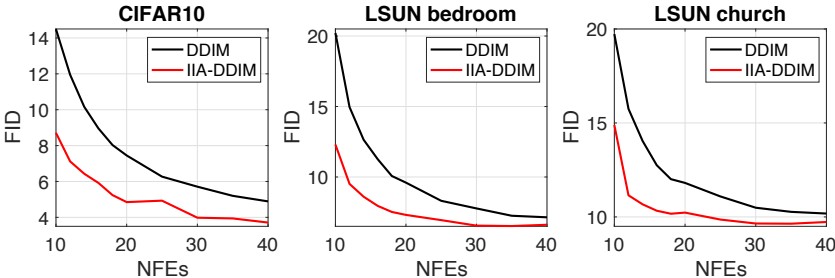

Figure 5: Sampling performance of DDIM and IIA-DDIM for conventional pre-trained models.

generated in FID and CLIP evaluation by using 20K different text prompts. All five methods share the same set of text prompts and the same seed of the random noise generator. The obtained images were resized to a size of $256 \times 256$ before computing the FID and CLIP scores. The tested NFEs were $\{10, 20, 30, 40\}$. The parameter $M$ in IIA-DDIM was set to $M = 10$, and the fine-grained timesteps were uniformly distributed within each time-interval. The set of quadratic functions for approximating the MSEs in IIA-DDIM and IIA-DPM-Solver++ were constructed by utilizing 20 different text-prompts from the validation set to compute the optimal stepsizes (see Fig. 7 for the optimal stepsizes $\{\beta_i^*\}$ in IIA-DDIM). Once the optimal stepsizes for each sampling method with a certain set of timesteps were obtained, 20K images were then generated accordingly.

Table 1 summarizes the FID and CLIP scores of the five methods. It is clear from the table that our two new methods IIA-DDIM and IIA-DPM-Solver++ perform significantly better than the original counterparts in terms of FID performance. The facts that the FIDs of DPM-Solver++ and PLMS are larger than that of DDIM at 10 NFEs might be because the gradient statistics of the classifier-free guided diffusion sampling are different from those of the conventional diffusion sampling. As a result, the stepsizes in front of the gradients in DPM-Solver++ and PLMS might not be optimal for the scenario of the classifier-free guided diffusion sampling when NFE is small. On the other hand, different stepsizes in IIA-DDIM and IIA-DPM-Solver++ are learned via MMSE for different pre-trained models no matter if it is classifier-free guided diffusion sampling or conventional sampling.

**Conventional DDIM sampling:** In the second experiment, we studied the performance gain of IIA-DDIM in comparison to DDIM. We tested three pre-trained models (see Table 11 in the appendix), one for a particular dataset: CIFAR10, LSUN bedroom, and LSUN church. The set-size $|\mathcal{B}|$ for approximating the expectation operation in (7) was set to 16, which is also the mini-batch size for sampling in the computation of the FID scores. The hyper-parameter $M$ in (7) was set to $M = 3$.

The performance results of DDIM and IIA-DDIM are shown in Fig. 5. It is seen that IIA-DDIM outperforms DDIM consistently for different NFEs and across different pre-trained models. The performance of IIA-SPNDM and IIA-IPNDM is shown in the Appendix G.

To summarize, in all our experiments, the IIA based ODE solvers improve the image quality remarkably when the NFE is small (i.e., less than 25). The performance gain of the IIA technique is obtained by improving the accuracy of the integration approximation per timestep via stepsize optimization. When the NFE is large, the integration approximation of existing ODE solvers is accurate in general, leaving little room to improve the accuracy to a large extent. Consequently, there is no significant improvement for large NFE when applying our IIA technique.

## 6 CONCLUSION

In this paper, we have proposed a new technique of improved integration approximation (IIA) to accelerate the diffusion-based sampling processes. In particular, we have proposed to introduce new stepsizes (coefficients) in front of certain quantities in existing popular ODE solvers in order to improve the accuracy of integration approximation. The stepsizes at timestep $t_i$ are determined by encouraging a coarse integration approximation over $[t_i, t_{i+1}]$ to get closer to a highly accurate integration approximation over the same time slot. The optimal stepsizes only need to be computed once and can then be stored and reused later on for extensive sampling. Extensive experiments confirm that the IIA technique is able to significantly improve the sampling quality of EDM, DDIM, and DPM-Solver++ when the NFE is small (e.g., less than 25). This can be explained by the fact that the integration approximation in the original method for small NFE is a rough estimate. The employment of IIA has significantly improved the accuracy of the integration approximation.

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

## A    UPDATE PROCEDURE OF BIIA-EDM

---

**Algorithm 2** BIIA-EDM as an extension of EDM in Karras et al. (2022)

---

1: **Input:**
2:    number of time steps $N, r = 1, \{\gamma_{ik}^*|k = 0, 1\}_{i=1}^{N-2}$  [ pre-computed values obtained via MMSE]
3: **Sample** $\boldsymbol{z}_0 \sim \mathcal{N}(\boldsymbol{0}, \alpha_{t_0}^2 \bar{\sigma}_{t_0}^2 \boldsymbol{I})$
4: **for** $i \in \{0, 1, \ldots, N-1\}$ **do**
5:     $\boldsymbol{d}_i \leftarrow \boldsymbol{d}(\boldsymbol{z}_i, t_i) = \left(\frac{\dot{\bar{\sigma}}_{t_i}}{\bar{\sigma}_{t_i}} + \frac{\dot{\alpha}_{t_i}}{\alpha_{t_i}}\right) \boldsymbol{z}_i - \frac{\dot{\bar{\sigma}}_{t_i} \alpha_{t_i}}{\bar{\sigma}_{t_i}} D_{\boldsymbol{\theta}}(\boldsymbol{z}_i; t_i)$
6:     $\tilde{\boldsymbol{z}}_{i+1} \leftarrow \boldsymbol{z}_i + (t_{i+1} - t_i)\boldsymbol{d}_i$
7:     **if** $\sigma_{t_{i+1}} \neq 0$ **then**
8:         $\boldsymbol{d}'_{i+1|i} \leftarrow \boldsymbol{d}(\tilde{\boldsymbol{z}}_{i+1}, t_{i+1}) = \left(\frac{\dot{\bar{\sigma}}_{t_{i+1}}}{\bar{\sigma}_{t_{i+1}}} + \frac{\dot{\alpha}_{t_{i+1}}}{\alpha_{t_{i+1}}}\right) \tilde{\boldsymbol{z}}_{i+1} - \frac{\dot{\bar{\sigma}}_{t_{i+1}} \alpha_{t_{i+1}}}{\bar{\sigma}_{t_{i+1}}} D_{\boldsymbol{\theta}}(\tilde{\boldsymbol{z}}_{i+1}; t_{i+1})$
9:         $\boldsymbol{z}_{i+1} \leftarrow \boldsymbol{z}_i + (t_{i+1} - t_i) \sum_{k=0}^{r} \gamma_{ik}^* \left(\frac{1}{2}\boldsymbol{d}_{i-k} + \frac{1}{2}\boldsymbol{d}'_{i-k+1|i-k}\right)$  [historical gradients are used]
10:    **end if**
11: **end for**
12: **Output:** $\boldsymbol{z}_N$

---

**Remark:** Sampling of Karras et al. (2022) is recovered when $\{\gamma_{i0}^* = 1\}_{i=0}^{N-2}$ and $\{\gamma_{ik}^* = 0|k \neq 0\}_{i=0}^{N-2}$.

## B    PROOF FOR LEMMA 1

Firstly, we rewrite the update expression for $\boldsymbol{z}_{i+1}$ in terms of $\boldsymbol{z}_i$, and the two estimators $\boldsymbol{D}_{\boldsymbol{\theta}}(\boldsymbol{z}_i, t_i)$ and $\boldsymbol{D}_{\boldsymbol{\theta}}(\tilde{\boldsymbol{z}}_{i+1}, t_{i+1})$:

$$
\begin{aligned}
\boldsymbol{z}_{i+1} &= \boldsymbol{z}_i + (t_{i+1} - t_i)(0.5\boldsymbol{d}_i + 0.5\boldsymbol{d}'_i) \\
&= \boldsymbol{z}_i + (t_{i+1} - t_i) \left( \frac{\boldsymbol{z}_i - \boldsymbol{D}_{\boldsymbol{\theta}(\boldsymbol{z}_i, t_i)}}{2t_i} + \frac{\tilde{\boldsymbol{z}}_{i+1} - \boldsymbol{D}_{\boldsymbol{\theta}}(\tilde{\boldsymbol{z}}_{i+1}, t_{i+1})}{2t_{i+1}} \right) \\
&= \boldsymbol{z}_i + (t_{i+1} - t_i)\frac{\boldsymbol{z}_i - \boldsymbol{D}_{\boldsymbol{\theta}}(\boldsymbol{z}_i, t_i)}{2t_i} \\
&\quad + (t_{i+1} - t_i)\frac{\boldsymbol{z}_i + (t_{i+1} - t_i)(\boldsymbol{z}_i - \boldsymbol{D}_{\boldsymbol{\theta}}(\boldsymbol{z}_i, t_i))/t_i - \boldsymbol{D}_{\boldsymbol{\theta}}(\tilde{\boldsymbol{z}}_{i+1}, t_{i+1})}{2t_{i+1}} \\
&\overset{\text{assume}}{=} \boldsymbol{z}_i + (t_{i+1} - t_i)\frac{\boldsymbol{z}_i - \bar{\boldsymbol{D}}_{\boldsymbol{\theta}}(\boldsymbol{z}_i, \tilde{\boldsymbol{z}}_{i+1})}{t_i}.
\end{aligned}
\tag{20}
$$

Next, we derive the expression for $\bar{\boldsymbol{D}}_{\boldsymbol{\theta}}(\boldsymbol{z}_i, \tilde{\boldsymbol{z}}_{i+1})$ in (20). To do so, we let

$$
\begin{aligned}
\frac{\boldsymbol{z}_i - \bar{\boldsymbol{D}}_{\boldsymbol{\theta}}(\boldsymbol{z}_i, \tilde{\boldsymbol{z}}_{i+1})}{t_i} &= \frac{\boldsymbol{z}_i - \boldsymbol{D}_{\boldsymbol{\theta}}(\boldsymbol{z}_i, t_i)}{2t_i} + \frac{\boldsymbol{z}_i + (t_{i+1} - t_i)(\boldsymbol{z}_i - \boldsymbol{D}_{\boldsymbol{\theta}(\boldsymbol{z}_i, t_i)})/t_i - \boldsymbol{D}_{\boldsymbol{\theta}}(\tilde{\boldsymbol{z}}_{i+1}, t_{i+1})}{2t_{i+1}} \\
\Leftrightarrow \boldsymbol{z}_i - \bar{\boldsymbol{D}}_{\boldsymbol{\theta}}(\boldsymbol{z}_i, \tilde{\boldsymbol{z}}_{i+1}) &= 0.5(\boldsymbol{z}_i - \boldsymbol{D}_{\boldsymbol{\theta}}(\boldsymbol{z}_i, t_i)) + \frac{t_i\boldsymbol{z}_i + (t_{i+1} - t_i)(\boldsymbol{z}_i - \boldsymbol{D}_{\boldsymbol{\theta}}(\boldsymbol{z}_i, t_i)) - t_i\boldsymbol{D}_{\boldsymbol{\theta}}(\tilde{\boldsymbol{z}}_{i+1}, t_{i+1})}{2t_{i+1}} \\
\Leftrightarrow -\bar{\boldsymbol{D}}_{\boldsymbol{\theta}}(\boldsymbol{z}_i, \tilde{\boldsymbol{z}}_{i+1}) &= -0.5\boldsymbol{D}_{\boldsymbol{\theta}}(\boldsymbol{z}_i, t_i) + \frac{-(t_{i+1} - t_i)\boldsymbol{D}_{\boldsymbol{\theta}(\boldsymbol{z}_i, t_i)} - t_i\boldsymbol{D}_{\boldsymbol{\theta}}(\tilde{\boldsymbol{z}}_{i+1}, t_{i+1})}{2t_{i+1}} \\
\Leftrightarrow -\bar{\boldsymbol{D}}_{\boldsymbol{\theta}}(\boldsymbol{z}_i, \tilde{\boldsymbol{z}}_{i+1}) &= -\boldsymbol{D}_{\boldsymbol{\theta}}(\boldsymbol{z}_i, t_i) + \frac{t_i(\boldsymbol{D}_{\boldsymbol{\theta}}(\boldsymbol{z}_i, t_i) - \boldsymbol{D}_{\boldsymbol{\theta}}(\tilde{\boldsymbol{z}}_{i+1}, t_{i+1}))}{2t_{i+1}} \\
\Leftrightarrow \bar{\boldsymbol{D}}_{\boldsymbol{\theta}}(\boldsymbol{z}_i, \tilde{\boldsymbol{z}}_{i+1}) &= \boldsymbol{D}_{\boldsymbol{\theta}}(\boldsymbol{z}_i, t_i) + \frac{t_i(\boldsymbol{D}_{\boldsymbol{\theta}}(\tilde{\boldsymbol{z}}_{i+1}, t_{i+1}) - \boldsymbol{D}_{\boldsymbol{\theta}}(\boldsymbol{z}_i, t_i))}{2t_{i+1}}.
\end{aligned}
\tag{21}
$$

Plugging (21) into (20), and rearranging the terms in the expression yields (17). The proof is complete.

## C    DESIGN OF IIA-DPM-SOLVER++ FOR CLASSIFIER-FREE GUIDED TEXT-TO-IMAGE GENERATION

In general, DPM-Solver has different implementations in the platform of StableDiffusion V2. The results in Table 1 were obtained by using the multi-step 2nd-order DPM-Solver++, which is the default setup in StableDiffusion. At each timestep $t_j$, the pre-trained DNN model produces an estimator $\hat{\boldsymbol{x}}_{\boldsymbol{\theta}}(\boldsymbol{z}_j, \phi = P, t_j)$ of the clean-image, where $P$ denotes the text prompt. In general, when $i > 0$, the diffusion state $\boldsymbol{z}_{i+1}$ is computed by making use of the two most recent clean-image estimators $\{\hat{\boldsymbol{x}}_{\boldsymbol{\theta}}(\boldsymbol{z}_j, \phi = P, t_j)|j = i - 1, i\}$ as well as the current state $\boldsymbol{z}_i$. For simplicity, let us denote the update expression of DPM-Solver++ for computing $\boldsymbol{z}_{i+1}$ at timestep $t_i$ as

$$\boldsymbol{z}_{i+1} = \Gamma_{t_i \to t_{i+1}}\left(\boldsymbol{z}_i, \{\hat{\boldsymbol{x}}_{\boldsymbol{\theta}}(\boldsymbol{z}_j, \phi = P, t_j)|j = i - 1, i\}\right). \tag{22}$$

Next, we consider refining the estimation for $\boldsymbol{z}_{i+1}$ in (22) by using the IIA technique. Similarly to the design of IIA-DDIM, we propose to compute $\boldsymbol{z}_{i+1}$ at timestep $t_i$ by introducing two additional quantities into (22), which takes the form of

$$\boldsymbol{z}_{i+1} = \Gamma_{t_i \to t_{i+1}}\left(\boldsymbol{z}_i, \{\hat{\boldsymbol{x}}_{\boldsymbol{\theta}}(\boldsymbol{z}_j, \phi = P, t_j)|j = i - 1, i\}\right) + \varphi_{i0}\boldsymbol{z}_i + \varphi_{i1}\hat{\boldsymbol{x}}_{\boldsymbol{\theta}}(\boldsymbol{z}_i, \phi = P, t_i). \tag{23}$$

To optimize the two stepsizes $(\varphi_{i0}, \varphi_{i1})$ in (23), we construct the following MSE function

$$
\begin{aligned}
&(\varphi_{i0}^*, \varphi_{i1}^*) \\
&= \arg\min \mathbb{E}\Big\|\Gamma_{t_i \to t_{i+1}}\left(\boldsymbol{z}_i, \{\hat{\boldsymbol{x}}_{\boldsymbol{\theta}}(\boldsymbol{z}_j, \phi = P, t_j)|j = i - 1, i\}\right) \\
&+ \varphi_{i0}\boldsymbol{z}_i + \varphi_{i1}\hat{\boldsymbol{x}}_{\boldsymbol{\theta}}(\boldsymbol{z}_i, \phi = P, t_i) \\
&- \boldsymbol{z}_i - \sum_{m=0}^{M-1}\Big(\boldsymbol{\Gamma}_{t_{i+\frac{m}{M}} \to t_{i+\frac{m+1}{M}}}\left(\boldsymbol{z}_{i+\frac{m}{M}}, \{\hat{\boldsymbol{x}}_{\boldsymbol{\theta}}(\boldsymbol{z}_j, \phi = P, t_j)|j = m - 1, m\}\right) - \boldsymbol{z}_{i+\frac{m}{M}}\Big)\Big\|^2, \tag{24}
\end{aligned}
$$

where the expectation is taken over the distribution of initial Gaussian noise $\boldsymbol{z}_{t_0} \sim \mathcal{N}(0, \sigma_T^2\boldsymbol{I})$ and the distribution of the text-prompt $P \sim p_{text}$. The summation from $m = 0$ to $m = M - 1$ in the RHS of (24) corresponds to applying the original DPM-Solver++ over a set of fine-grained timeslots within $[t_i, t_{i+1}]$. We use $\mathcal{C}$ to denote a set of finite pairs of $(\boldsymbol{z}_{t_0}, P)$. The MSE in (24) can then be approximated by using the finite set $\mathcal{C}$. In our experiment for both IIA-DDIM and IIA-DPM-Solver in the task of text-to-image generation, the set-size of $\mathcal{C}$ was set to 20. The optimal solution $(\varphi_{i0}^*, \varphi_{i1}^*)$ can be computed by solving a quadratic optimization problem based on $\mathcal{C}$. Once the optimal stepsizes $(\varphi_{i0}^*, \varphi_{i1}^*)_{i=1}^{N-1}$ are computed for the first time, they are then utilized for generating 20K images in FID and CLIP evaluation by feeding 20K different text-prompts from COCO2014 validation set.

## D    ABLATION STUDY OF IIA-BASED ODE SOLVERS

In this section, we first present the setup of the hyperparameters in IIA when performing MMSE. We then study the performance of BIIA-EDM and IIA-EDM for different $r$ values. After that, we examine how the parameter $M$ affects the performance of IIA-EDM with 11 and 19 NFEs. Finally, we consider the performance of IIA-DDIM by optimizing one coefficient per timestep instead of two coefficients per step as being studied in (6)-(7).

### D.1    HYPER-PARAMETERS OF IIA WHEN PERFORMING MMSE

The reason we select $r = 1$ or $r = 0$ in Table 2 for BIIA-EDM and IIA-EDM is because the ablation study in Subsection D.2 over CIFAR10 shows that the setup $r = 2$ does not generally improve the FID performance. It is also noted in Table 2 that only for BIIA-EDM over FFHQ64, the setup $r = 0$ is being employed instead of $r = 1$. The ablation study in Table 3 shows that the performance of $r = 0$ is slightly better than that of $r = 1$ in BIIA-EDM over FFHQ64.

For the experiment of IIA-DDIM, IIA-SPNDM, and IIA-IPDNM, the hyper-parameter $M$ was set to $M = 3$. Similarly, the fine-grained timestep $t_{i+\frac{m}{M}}$ was computed as $t_{i+\frac{m}{M}} = t_i + \frac{(t_{i+1} - t_i)m}{M}$.

### D.2    ABLATION STUDY FOR BIIA-EDM AND IIA-EDM WITH DIFFERENT $r$ VALUES

In this section, we perform an ablation study for BIIA-DDIM and IIA-EDM with $r = 1$ and $r = 2$. The parameter $M$ was set to $M = 3$, which is the same as the setup for obtaining Figure 3.

Table 2: Parameter-setups when performing MMSE in BIIA-EDM and IIA-EDM. The four pre-trained models below were downloaded from the official open source repository of the work Karras et al. (2022). The fine-grained timesteps $\{t_{i+\frac{m}{M}}\}_{m=0}^{M}$ were uniformly distributed within each time slot $[t_i, t_{i+1}]$. In particular, $t_{i+\frac{m}{M}}$ was computed as $t_{i+\frac{m}{M}} = t_i + \frac{(t_{i+1}-t_i)m}{M}$.

| pre-trained models | BIIA-EDM | IIA-EDM |
|---|---|---|
| edm-cifar10-32x32-cond-vp.pkl | $(M, r) = (3, 1)$ | $(M, r) = (3, 1)$ |
| edm-ffhq-64x64-uncond-vp.pkl | $(M, r) = (3, 0)$ | $(M, r) = (3, 1)$ |
| edm-afhqv2-64x64-uncond-vp.pkl | $(M, r) = (3, 1)$ | $(M, r) = (3, 1)$ |
| edm-imagenet-64x64-cond-adm.pkl | $(M, r) = (3, 1)$ | $(M, r) = (3, 1)$ |

Table 3: FID comparison of the setups $r = 1$ and $r = 0$ in BIIA-EDM for FFHQ64. The parameter $M$ was set to $M = 3$.

| NFEs | 11 | 13 | 15 | 19 |
|---|---|---|---|---|
| BIIA-EDM ($r = 1$) | 16.01 | 9.24 | 6.25 | 4.02 |
| BIIA-EDM ($r = 0$) | 12.78 | 7.33 | 5.09 | 3.66 |
| EDM | | 29.34 | 15.87 | 9.85 | 5.26 |

Table 4 and 5 summarize the averaged residual errors and the FID scores of the two methods, respectively. By inspection of the FID scores, it is seen that as $r$ increases from $r = 1$ to $r = 2$, the performance for each sampling method is slightly degraded in most cases. On the other hand, as $r$ increases, the averaged residual errors for each method in Table 4 slightly decrease. The above analysis is a bit counter-intuitive. It implies that when we introduce more gradient vectors in IIA-EDM or BIIA-EDM, it reduces the averaged residual error but does not always improve the FID score. This suggests that in addition to the residual errors of the MSEs, there are some unknown factors that affect the image sampling quality.

It is noted from Table 5 that for each sampling method, the differences of the FID scores for $r = 1$ and $r = 2$ are very small. In practice, it does not matter much regarding the FID scores if one chooses $\gamma = 1$ or $= 2$. However, the setup $\gamma = 2$ would incur additional memory consumption. It is thus recommended to use $\gamma = 1$ as least for CFIAR10 in Table 5.

Table 4: Averaged residual errors of BIIA-EDM and IIA-EDM with $r = 1$ and $r = 2$.

| | NFEs | BIIA-EDM ($r = 2$) | BIIA-EDM ($r = 1$) | IIA-EDM ($r = 2$) | IIA-EDM ($r = 1$) | EDM |
|---|---|---|---|---|---|---|
| CIFAR 10 | 11 | 4.31 | 4.93 | 2.87 | 2.98 | 17.33 |
| | 13 | 1.75 | 2.08 | 1.07 | 1.12 | 6.22 |
| | 15 | 0.84 | 1.00 | 0.52 | 0.54 | 2.58 |
| | 19 | 0.24 | 0.28 | 0.14 | 0.15 | 0.61 |

Table 5: FID evaluation of BIIA-EDM and IIA-EDM with $r = 1$ and $r = 2$.

| | NFEs | BIIA-EDM ($r = 2$) | BIIA-EDM ($r = 1$) | IIA-EDM ($r = 2$) | IIA-EDM ($r = 1$) | EDM |
|---|---|---|---|---|---|---|
| CIFAR 10 | 11 | **7.81** | 7.97 | 5.01 | **4.90** | 14.42 |
| | 13 | 4.01 | **3.91** | 2.96 | **2.89** | 6.79 |
| | 15 | 2.87 | **2.76** | 2.37 | **2.33** | 4.31 |
| | 19 | 2.20 | **2.16** | 2.06 | **2.04** | 2.54 |

## D.3 ABLATION STUDY OF IIA-EDM FOR DIFFERENT $M$ VALUES

In this section, we study how the parameter $M$ affects the performance of IIA-EDM over CIFAR10. The tested $M$ values were $M \in \{3, 10, 20, 50\}$. Two NFEs were considered in the ablation study, which were 11 and 19.

The results are summarized in Table 6. It is clear from the table that the one-time computational overhead increases linearly as $M$ increases from 3 to 50. When the $M$ value is larger than 10, the obtained FID scores and the averaged residual errors are roughly the same. This suggests that in practice, a reasonably large $M$ value is sufficient when applying the IIA technique. For the results in Table 6, $M = 10$ is sufficiently large.

Table 6: Comparison of FID, computational overhead (in seconds), and averaged residual errors for different $M$ values in IIA-EDM over CIFAR10. The parameter $r$ was set to $r = 1$.

|  | $M$ | 3 | 10 | 20 | 50 |
|---|---|---|---|---|---|
| IIA-EDM (11 NFEs) | FID | 4.90 | 3.76 | 3.71 | 3.70 |
| IIA-EDM (11 NFEs) | computational overhead (s) | 17.6 | 46.8 | 88.0 | 210.5 |
| IIA-EDM (11 NFEs) | averaged residual errors | 2.98 | 3.32 | 3.35 | 3.36 |
| IIA-EDM (19 NFEs) | FID | 2.04 | 2.10 | 2.11 | 2.11 |
| IIA-EDM (19 NFEs) | computational overhead (s) | 34.1 | 91.9 | 171.8 | 418.9 |
| IIA-EDM (19 NFEs) | averaged residual errors | 0.146 | 0.179 | 0.181 | 0.182 |

### D.4 ABLATION STUDY OF IIA-DDIM BY OPTIMIZING ONE COEFFICIENT PER-TIMESTEP

In Section 3, we have presented IIA-DDIM for both conventional diffusion sampling and text-to-image generation. For the case of conventional diffusion sampling, we optimize two stepsizes in front of two quantities for IIA-DDIM as shown in (6). In this section, we consider the sampling performance of IIA-DDIM that involves only one quantity in terms of the difference of the estimated clean data. In particular, the update expression for $z_{i+1}$ is computed as

$$z_{i+1} = \overbrace{\alpha_{t_{i+1}} \hat{x}(z_i, t_i) + \sigma_{t_{i+1}} \hat{\epsilon}_\theta(z_i, t_i)}^{\Phi_{t_i \to t_{i+1}}(z_i, t_i)} + \phi_i^* \overbrace{(\hat{x}(z_i, t_i) - \hat{x}(z_{i-1}, t_{i-1}))}^{\text{additional quantity}}, \quad (25)$$

where the optimal coefficient $\phi_i^*$ can be obtained by MMSE. We omit the details for computing $\phi_i^*$ to avoid redundant derivation.

**Experimental comparison:** Table 7 summarizes the averaged residual errors of the associated MSEs for DDIM, and IIA-DDIM with one quantity and two quantities. As we explained earlier, IIA-DDIM with one quantity has the update expression (25) while IIA-DDIM with two quantities has the update expression (6). It is clear from the table that IIA-DDIM with two quantities produces the smallest residual error for each number of timesteps. The residual errors of IIA-DDIM with one quantity are also consistently smaller than those of DDIM. By following the design principle of IIA in Section 3, both versions of IIA-DDIM are good sampling candidates.

Table 8 below summarizes the obtained FID scores for the three sampling methods in Table 7. It is clear from the table that both versions of IIA-DDIM perform much better than DDIM. Furthermore, IIA-DDIM with two quantities outperforms IIA-DDIM with one quantity. The above results are consistent with the behaviors of averaged residual errors of the three methods in Table 7.

Table 7: Averaged residual errors of IIA-DDIM with one quantity (see (6)) and two quantities (see (25)) for conventional diffusion sampling (see also Fig. 5). The residual error of DDIM is also included as a reference.

|  | timesteps | IIA-DDIM (one quantity) | IIA-DDIM (two quantities) | DDIM |
|---|---|---|---|---|
| CIFAR10 | 10 | 8.20e-2 | **6.60e-2** | 1.73e-1 |
|  | 20 | 8.0e-3 | **4.50e-3** | 2.21e-2 |
|  | 40 | 8.50e-4 | **3.76e-4** | 3.10e-3 |

**Study of the two quantities :** We now show that the difference $\hat{x}(z_i, t_i) - \hat{x}(z_{i-1}, t_{i-1})$ of the estimated clean data cannot be represented by using the difference of the estimated Gaussian noises

Table 8: FID evaluation of IIA-DDIM with one quantity (see (6)) and two quantities (see (25)) for conventional diffusion sampling (see also Fig. 5). The performance of DDIM is also included as a reference.

| | timesteps | IIA-DDIM (one quantity) | IIA-DDIM (two quantities) | DDIM |
|---|---|---|---|---|
| CIFAR10 | 10 | 9.50 | **8.66** | 14.38 |
| | 20 | 5.43 | **4.85** | 7.51 |
| | 40 | 3.97 | **3.68** | 4.95 |

$\hat{\epsilon}_{\boldsymbol{\theta}}(\boldsymbol{z}_i, t_i) - \hat{\epsilon}_{\boldsymbol{\theta}}(\boldsymbol{z}_{i-1}, t_{i-1})$:

$$\hat{\boldsymbol{x}}(\boldsymbol{z}_i, t_i) - \hat{\boldsymbol{x}}(\boldsymbol{z}_{i-1}, t_{i-1})$$
$$= \frac{\boldsymbol{z}_i - \sigma_{t_i}\hat{\epsilon}_{\boldsymbol{\theta}}(\boldsymbol{z}_i, t_i)}{\alpha_{t_i}} - \frac{\boldsymbol{z}_{i-1} - \sigma_{t_{i-1}}\hat{\epsilon}_{\boldsymbol{\theta}}(\boldsymbol{z}_{i-1}, t_{i-1})}{\alpha_{t_{i-1}}}$$
$$= \left(\frac{\boldsymbol{z}_i}{\alpha_{t_i}} - \frac{\boldsymbol{z}_{i-1}}{\alpha_{t_{i-1}}}\right) - \left(\frac{\sigma_{t_i}\hat{\epsilon}_{\boldsymbol{\theta}}(\boldsymbol{z}_i, t_i)}{\alpha_{t_i}} - \frac{\sigma_{t_{i-1}}\hat{\epsilon}_{\boldsymbol{\theta}}(\boldsymbol{z}_{i-1}, t_{i-1})}{\alpha_{t_{i-1}}}\right), \tag{26}$$

where in general $\frac{\sigma_{t_i}}{\alpha_{t_i}} \neq \frac{\sigma_{t_{i-1}}}{\alpha_{t_{i-1}}}$. As a result, one can conclude from the above derivation that the additional two quantities in (6) cannot be reduced to one quantity together with a linear combination of $\boldsymbol{z}_i$ and $\boldsymbol{z}_{i-1}$.

## E SAMPLING TIME AND COMPUTATIONAL OVERHEAD OF IIA-EDM

Table 9: Comparison of sampling time between EDM and IIA-EDM (in seconds) over a GPU (NVIDIA RTX 2080Ti). The batchsize was set to 200. See Table 10 for the computational overhead of IIA-EDM.

| | NFEs | 11 | 13 | 15 | 17 | 19 | 21 | 23 |
|---|---|---|---|---|---|---|---|---|
| CIFAR10 | EDM | 5.1 | 6.1 | 7.1 | 8.2 | 9.2 | 10.3 | 11.4 |
| | IIA-EDM | 5.2 | 6.3 | 7.3 | 8.4 | 9.3 | 10.4 | 11.4 |
| FFHQ | EDM | 12.4 | 15.0 | 17.4 | 20.0 | 22.4 | 24.9 | 27.3 |
| | IIA-EDM | 12.6 | 15.2 | 17.5 | 20.0 | 22.5 | 25.0 | 27.4 |
| AFHQV2 | EDM | 12.6 | 15.0 | 17.5 | 19.9 | 22.3 | 24.9 | 27.4 |
| | IIA-EDM | 12.7 | 15.1 | 17.6 | 20.0 | 22.4 | 24.9 | 27.5 |

Table 10: One-time computational overhead (in seconds) of IIA-EDM for computing the optimal coefficients via MMSE.

| NFEs | 11 | 13 | 15 | 17 | 19 | 21 | 23 |
|---|---|---|---|---|---|---|---|
| CIFAR10 | 17.6 | 21.9 | 25.9 | 30.0 | 34.1 | 37.3 | 41.8 |
| FFHQ | 42.8 | 52.6 | 62.0 | 71.9 | 80.3 | 91.6 | 102.1 |
| AFHQV2 | 42.7 | 52.3 | 62.2 | 72.4 | 82.0 | 92.0 | 101.8 |

The GPU (NVIDIA RTX 2080Ti) was utilized for measuring the processing time (in seconds). The hyper-parameter $(|\mathcal{B}|, M, r)$ was set to $(M, r) = (200, 3, 1)$ as in the paper, where $|\mathcal{B}|$ denotes the number of samples in set $\mathcal{B}$ of initial noise vector $\boldsymbol{z}_{t_0}$. $r = 1$ refers to the case that only the gradient of the most recent time step is being utilized in IIA-EDM, which should not take much memory space.

## F TESTED PRE-TRAINED MODELS FOR IIA-DDIM, IIA-SPNDM AND IIA-IPNDM

Table 11: Tested pre-trained models in Fig. 5 and Fig. 6

| |
|---|
| 1.ddim_cifar10.ckpt |
| 2.ddim_lsun_bedroom.ckpt |
| 3.ddim_lsun_church.ckpt |
| (from `https://github.com/luping-liu/PNDM`) |

# G   PERFORMANCE OF IIA-SPNDM AND IIA-IPNDM

## G.1   DESIGN OF IIA-SPNDM

IIA-SPNDM is designed to solve a variance-preserving (VP) ODE (i.e., $\sigma_t = \sqrt{1 - \alpha_t^2}$ in (2)) by following a similar procedure for IIA-DDIM presented in Section 3. We summarize the sampling procedure of IIA-SPNDM in Alg. 3. The only difference between IIA-SPNDM and SPNDM is the computation of $z_{i+1}$ for $i = 1, \ldots, N - 1$, where two additional terms are introduced for better integration approximation. The two coefficients $\varphi_{i0}^*$ and $\varphi_{i01}^*$ in Alg. 3 can, in principle, be computed by performing the following MMSE

$$(\varphi_{i0}^*, \varphi_{i1}^*) = \arg\min \mathbb{E}_{\boldsymbol{z}_{t_0} \sim \mathcal{N}(0, \sigma_T^2 \boldsymbol{I})} \Big\| \boldsymbol{\Psi}_{t_i \to t_{i+1}}(\boldsymbol{z}_i, t_i) + \varphi_{i0}(\hat{\boldsymbol{x}}_{[i:i-1]} - \hat{\boldsymbol{x}}_{[i-1:i-2]})$$
$$+ \varphi_{i1}(\tilde{\boldsymbol{\epsilon}}_{[i:i-1]} - \tilde{\boldsymbol{\epsilon}}_{[i-1:i-2]}) - \sum_{m=0}^{M-1} \boldsymbol{\Psi}_{t_{i+\frac{m}{M}} \to t_{i+\frac{m+1}{M}}}(\boldsymbol{z}_{i+\frac{m}{M}}, t_{i+\frac{m}{M}}) \Big\|^2, \qquad (27)$$

where $\boldsymbol{\Psi}_{t_i \to t_{i+1}}(\boldsymbol{z}_i, t_i)$ represents the update expression of SPNDM over the time interval $[t_i, t_{i+1}]$, given by

$$\boldsymbol{\Psi}_{t_i \to t_{i+1}}(\boldsymbol{z}_i, t_i) = \alpha_{i+1} \overbrace{(\boldsymbol{z}_i - \sqrt{1 - \alpha_i^2}\tilde{\boldsymbol{\epsilon}}_{[i:i-1]})/\alpha_i}^{\hat{\boldsymbol{x}}_{[i:i-1]}} + \sqrt{1 - \alpha_{i+1}^2}\tilde{\boldsymbol{\epsilon}}_{[i:i-1]}, \qquad (28)$$

and the summation $\sum_{m=0}^{M-1} \boldsymbol{\Psi}_{t_{i+\frac{m}{M}} \to t_{i+\frac{m+1}{M}}}(\boldsymbol{z}_{i+\frac{m}{M}}, t_{i+\frac{m}{M}})$ in (27) provides a highly accurate integration approximation by applying SPNDM over a fine-grained set of timesteps within the time interval $[t_i, t_{i+1}]$. When the two coefficients are manually set to $(\varphi_{i0}^*, \varphi_{i01}^*) = (0, 0)$ for all $i$, IIA-SPNDM reduces to SPNDM.

From Alg. 3, we observe that the method SPNDM or IIA-SPNDM exploits 2nd order polynomial of the estimated Gaussian noises $\{\hat{\boldsymbol{\epsilon}}_{\boldsymbol{\theta}}(\boldsymbol{z}_{i-j}, i - j)\}_{j=0}^1$ in estimation of $\boldsymbol{z}_{i+1}$ at timestep $i > 0$. The coefficients $(3/2, -1/2)$ of the polynomial are fixed across different timesteps.

---

**Algorithm 3** Sampling of IIA-SPNDM

---

**Input:** $z_0 \sim \mathcal{N}(0, I)$, $\{\varphi_{i0}^*, \varphi_{i1}^*\}_{i=1}^{N-1}$

**for** $i = 0$ **do**

$(a)$ $\begin{cases} z_{i+1} = \frac{\alpha_{i+1}}{\alpha_i}\left(z_i - \sqrt{1 - \alpha_i^2}\hat{\epsilon}_\theta(z_i, i)\right) + \sqrt{1 - \alpha_{i+1}^2}\hat{\epsilon}_\theta(z_i, i) \\ \hat{\epsilon}_{[i+1:i]} = \frac{1}{2}(\hat{\epsilon}_\theta(z_i, i) + \hat{\epsilon}_\theta(z_{i+1}, i+1)) \\ \hat{x}_i = (z_i - \sqrt{1 - \alpha_i^2}\hat{\epsilon}_{[i+1:i]})/\alpha_i \\ z_{i+1} = \alpha_{i+1}\hat{x}_i + \sqrt{1 - \alpha_{i+1}^2}\hat{\epsilon}_{[i+1:i]} \end{cases}$

**end for**

Denote $\hat{x}_{[0:-1]} = \hat{x}_0$

**for** $i = 1 \ldots, N - 1$ **do**

$(b)$ $\begin{cases} \tilde{\epsilon}_{[i:i-1]} = \frac{1}{2}(3\hat{\epsilon}_\theta(z_i, i) - \hat{\epsilon}_\theta(z_{i-1}, i-1)) \\ \hat{x}_{[i:i-1]} = (z_i - \sqrt{1 - \alpha_i^2}\tilde{\epsilon}_{[i:i-1]})/\alpha_i \\ z_{i+1} = \alpha_{i+1}\hat{x}_{[i:i-1]} + \sqrt{1 - \alpha_{i+1}^2}\tilde{\epsilon}_{[i:i-1]} + \varphi_{i0}^*(\hat{x}_{[i:i-1]} - \hat{x}_{[i-1:i-2]}) \\ \qquad\qquad + \varphi_{i1}^*(\tilde{\epsilon}_{[i:i-1]} - \tilde{\epsilon}_{[i-1:i-2]}) \end{cases}$

**end for**

**output:** $z_N$

---

* The update for $z_1$ in $(a)$ is referred to as pseudo improved Euler step in Liu et al. (2022).
* The update for $z_{i+1}$ in $(b)$ is referred to as pseudo linear multi step in Liu et al. (2022).
* IIA-SPNDM reduces to SPNDM when $\{\varphi_{i0}^* = 0, \varphi_{i1}^* = 0\}_{i=1}^{N-1}$.

---

### G.2 SAMPLING PROCEDURE OF IIA-IPNDM

In brief, IPNDM is a 4th-order ODE solver Zhang & Chen (2022) as an extension of the PNDM method Liu et al. (2022). At timestep $i$, the four most recent estimated Gaussian noises $\{\epsilon_\theta(z_{i-j}, i - j)\}_{j=0}^3$ are linearly combined to produce a more reliable estimated Gaussian noise $\tilde{\epsilon}_{\theta,i}$. IPNDM then utilizes $\tilde{\epsilon}_{\theta,i}$ and $z_i$ to compute the next diffusion state $z_{i+1}$.

We extend IPNDM to obtain IIA-IPDNM, aiming to find out if the IIA technique can assist the sampling performance of IPNDM. The sampling procedure of IIA-PNDM is summarized in Alg. 4. The two coefficients $(\varphi_{i0}^*, \varphi_{i1}^*)$ at iteration $i$ are pre-determined by the IIA technique via solving a quadratic optimisation, which is constructed in a similar way as (27). We omit the details here.

---

**Algorithm 4** Sampling of IIA-IPNDM

---

**Input:** $z_0 \sim \mathcal{N}(0, I)$, $\{\varphi_{i0}^*, \varphi_{i1}^*\}_{i=1}^{N-1}$

**for** $i = 0$ **do**

$\quad \hat{x}_i = (z_i - \sqrt{1 - \alpha_i^2}\epsilon_\theta(z_i, i))/\alpha_i$

$\quad z_{i+1} = \alpha_{i+1}\hat{x}_i + \sqrt{1 - \alpha_{i+1}^2}\epsilon_\theta(z_i, i)$

**end for**

**for** $i = 1 \ldots, N - 1$ **do**

$\quad$ **if** $i = 1$ **then**

$\quad\quad \tilde{\epsilon}_{\theta,i} = (3\epsilon_\theta(z_i, i) - \epsilon_\theta(z_{i-1}, i-1))/2$

$\quad$ **else if** small $i = 2$ **then**

$\quad\quad \tilde{\epsilon}_{\theta,i} = (23\epsilon_\theta(z_i, i) - 16\epsilon_\theta(z_{i-1}, i-1) + 5\epsilon_\theta(z_{i-2}, i-2))/12$

$\quad$ **else**

$\quad\quad \tilde{\epsilon}_{\theta,i} = (55\epsilon_\theta(z_i, i) - 59\epsilon_\theta(z_{i-1}, i-1) + 37\epsilon_\theta(z_{i-2}, i-2) - 9\epsilon_\theta(z_{i-3}, i-3))/24$

$\quad$ **end if**

$\quad \hat{x}_i = (z_i - \sqrt{1 - \alpha_i^2}\tilde{\epsilon}_{\theta,i})/\alpha_i$

$\quad z_{i+1} = \alpha_{i+1}\hat{x}_i + \sqrt{1 - \alpha_{i+1}^2}\tilde{\epsilon}_{\theta,i} + \varphi_{i0}^*(\hat{x}_i - \hat{x}_{i-1}) + \varphi_{i1}^*(\tilde{\epsilon}_{\theta,i} - \tilde{\epsilon}_{\theta,i-1})$

**end for**

**output:** $z_N$

---

* IIA-IPNDM reduces to IPNDM when $\{\varphi_{i0}^* = 0, \varphi_{i1}^* = 0\}_{i=1}^{N-1}$.

---

---

**Algorithm 5** IPNDM

---

**Input:** $z_N \sim \mathcal{N}(\mathbf{0}, \mathbf{I})$
**for** $i = N$ **do**
    $\hat{x}_i = (z_i - \sqrt{1 - \alpha_i^2} \epsilon_\theta(z_i, i))/\alpha_i$
    $z_{i-1} = \alpha_{i-1}\hat{x}_i + \sqrt{1 - \alpha_{i-1}^2}\epsilon_\theta(z_i, i)$
**end for**
**for** $i = N - 1 \ldots, 0$ **do**
    **if** $i = N - 1$ **then**
        $\tilde{\epsilon}_{\theta,i} = (3\epsilon_\theta(z_i, i) - \epsilon_\theta(z_{i+1}, i+1))/2$
    **else if** small $i = 2$ **then**
        $\tilde{\epsilon}_{\theta,i} = (23\epsilon_\theta(z_i, i) - 16\epsilon_\theta(z_{i+1}, i+1) + 5\epsilon_\theta(z_{i+2}, i+2))/12$
    **else**
        $\tilde{\epsilon}_{\theta,i} = (55\epsilon_\theta(z_i, i) - 59\epsilon_\theta(z_{i+1}, i+1) + 37\epsilon_\theta(z_{i+2}, i+2) - 9\epsilon_\theta(z_{i+3}, i+3))/24$
    **end if**
    $\hat{x}_i = (z_i - \sqrt{1 - \alpha_i^2}\tilde{\epsilon}_{\theta,i})/\alpha_i$
    $z_{i-1} = \alpha_{i-1}\hat{x}_i + \sqrt{1 - \alpha_{i-1}^2}\tilde{\epsilon}_{\theta,i}$
**end for**
**output:** $z_0$

---

### G.3 PERFORMANCE COMPARISON

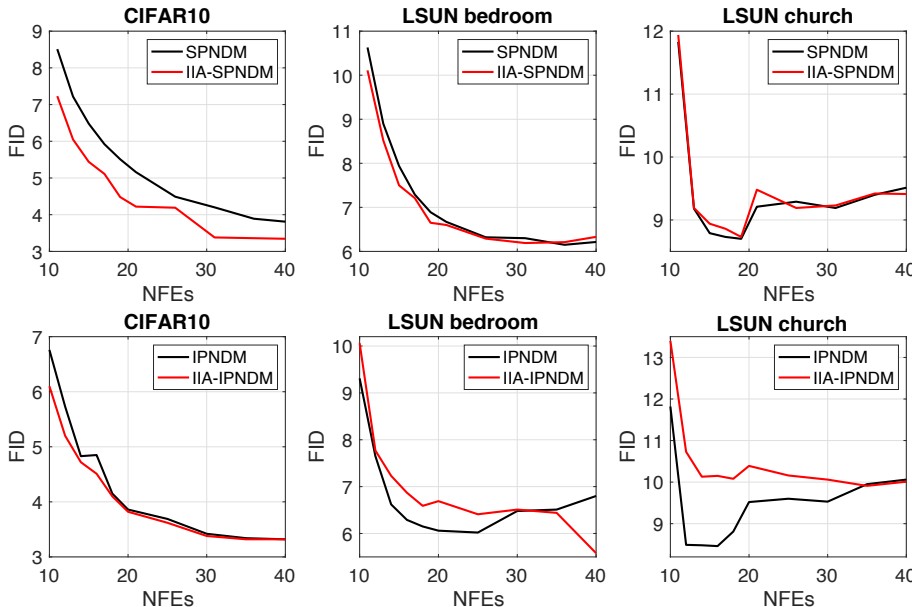

Figure 6: Performance comparison of four sampling methods.

In this experiment, we investigate the sampling performance of four methods: SPNDM, IIA-SPNDM, IPDNM, and IIA-IPNDM. The experimental setup follows that of IIA-DDIM in Subsection 5.2 and Section F. The tested pre-trained models are listed in Table F.

Fig. 6 summarizes the performance of the four sampling methods for small NFEs. It is clear that IIA-SPNDM outperforms SPNDM for CIFAR10. For LSUN-bedroom and LSUN-church, the performance of IIA-SPNDM and SPNDM is almost identical.

Next, we consider the performance of IIA-IPNDM and IPNDM. It is seen from the figure that for CIFAR10, IIA-IPNDM produces slightly better performance. However, for LSUN-bedroom and LSUN-church, the IIA technique does not help the sampling procedure of IPNDM. This can be explained by the fact that for LSUN-bedroom and LSUN-church, the FID score of IPNDM first decreases and then quickly increases in the NFE range of $[15 - 40]$, which is undesirable. This implies that as the NFE increases from 15 to 40, the accuracy of the integration approximation of

IPNDM may not be monotonically increasing. We note that the IIA technique implicitly assumes that a highly accurate integration approximation for each timeslot $[t_i, t_{i+1}]$ can be obtained by performing IPNDM over a set of fine-grained timesteps within $[t_i, t_{i+1}]$. Our above analysis suggests that the assumption of the IIA technique might be violated in the NFE range of [15, 40] for LSUN-bedroom and LSUN-church.

To summarize, the IIA technique improves the sampling performance of SPNDM and IPNDM for certain pre-trained models when the FID score decreases as the NFE increases. On the other hand, the IIA technique does not help with the sampling performance of SPNDM and IPNDM for those pre-trained models where the FID score first decreases and then quickly increases as the NFE increases.

## H    EXPERIMENTS ON TEXT-TO-IMAGE GENERATION

In our experiment, the pre-trained model used for text-to-image generation over StableDiffusion V2 is "v2-1_512-ema-pruned.ckpt". The three reference methods DDIM, PLMS and DPM-Solver++ are implemented by StableDiffusion V2 itself.

Fig. 7 below summarizes the obtained optimal $\beta$ values (see (8)) in IIA-DDIM for the text-to-image generation task. As can be seen, for each NFE scenario, the optimal $\beta$ values are different across different timestep indices. As $t_i$ approaches to $t_N = 0$, the optimal $\beta$ parameter increases. Furthermore, as NFE increases from 10 to 40, the average of the beta values decreases. From the above analysis, we can conclude that it is time-consuming to manually tune the parameter $\beta$.

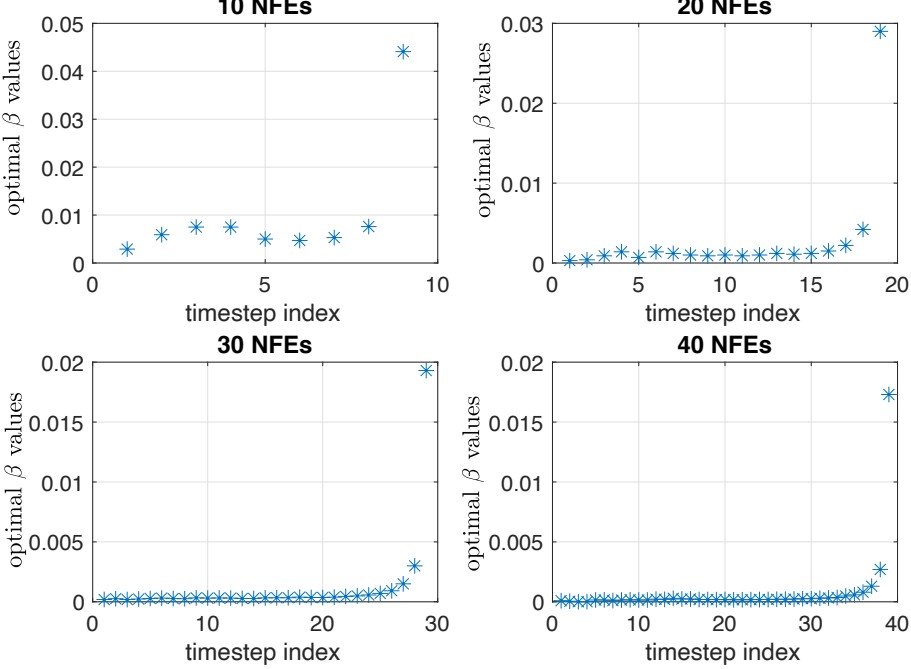

Figure 7: Optimal $\beta$ values in IIA-DDIM for classifier-free guided text-to-image sampling.

# I  ADDITIONAL IMAGE COMPARISONS

Table 12: text-prompts in Fig. 1, 8, and 9.

| | |
|---|---|
| (a) | A bench sitting along side of river next to tree |
| (b) | A blonde boy stands looking happily at the camera |
| (c) | A double decker bus is moving along a stretch of road |
| (d) | A large black bear standing in a forest |
| (e) | A blue and light green bus parked at a terminal |
| (f) | Two cats sleep together in a open case |
| (g) | a black bench and a green and blue bottle |
| (h) | The sheep graze and eat in a city field |
| (i) | A sheep with horns in a grassy green field |
| (j) | Flowers in a vase on top of a wooden table |
| (k) | A large white bear standing near a rock |
| (l) | A man in glasses wearing a suit and tie |
| (m) | A cat that is looking at a dog |
| (n) | A man dressed for the snowy mountain looks at the camera |
| (o) | A bird standing alone in the water looking |

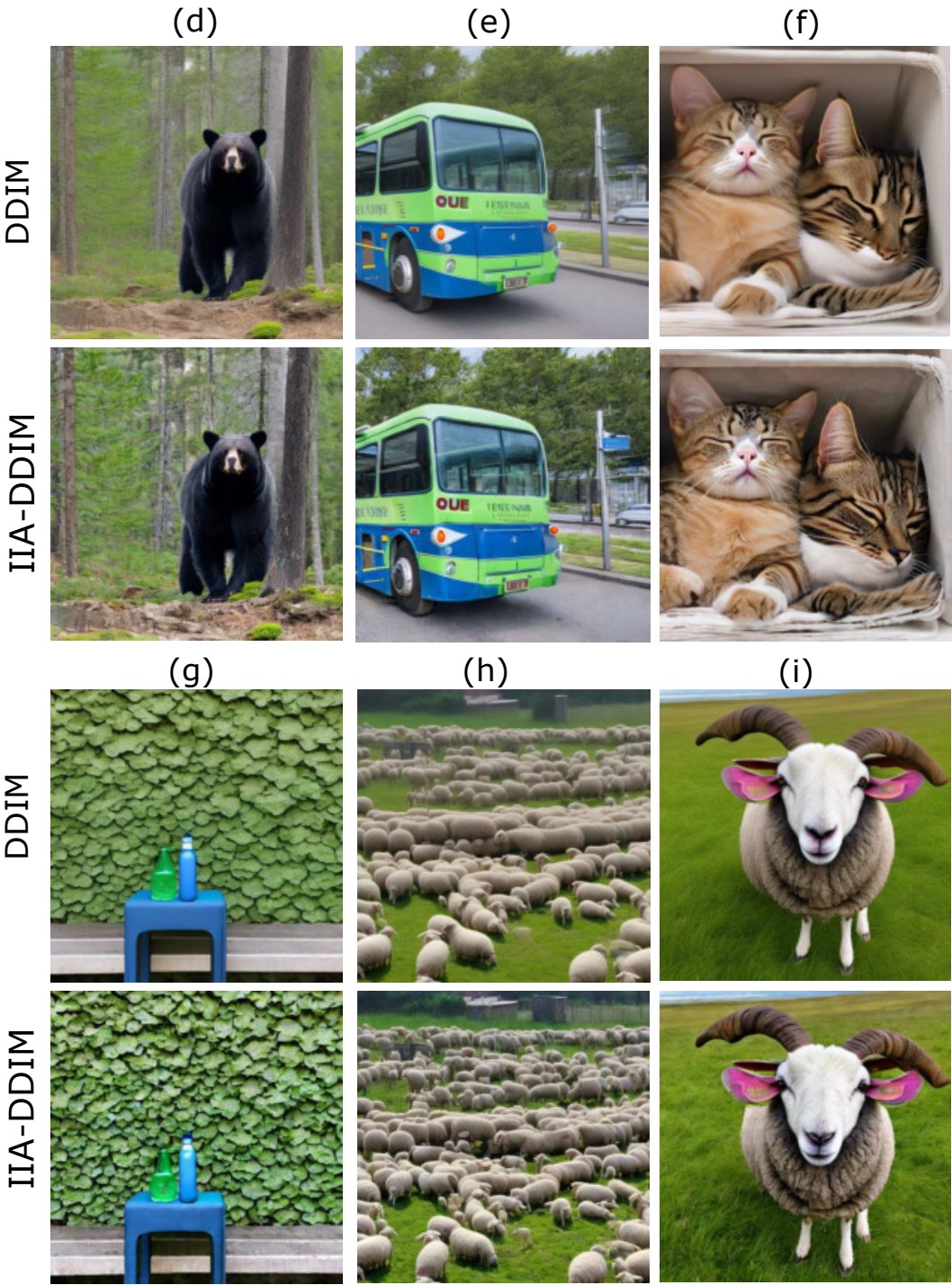

Figure 8: Comparison of images generated by DDIM and IIA-DDIM at 10 timesteps over StableDiffusion V2. See Table 12 for input texts.

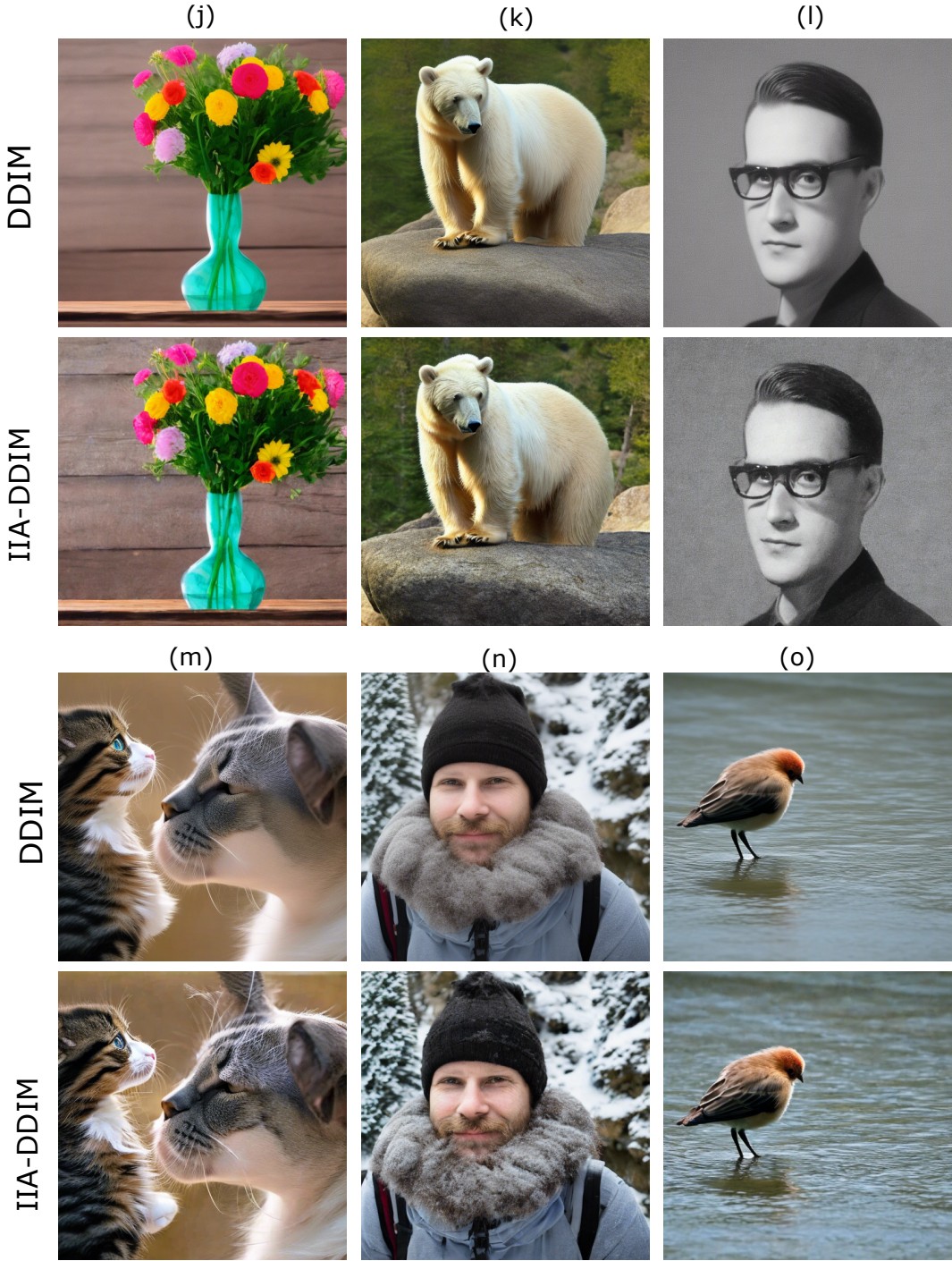

Figure 9: Comparison of images generated by DDIM and IIA-DDIM at 10 timesteps over StableDiffusion V2. See Table 12 for input texts.

EDM                                    IIA-EDM

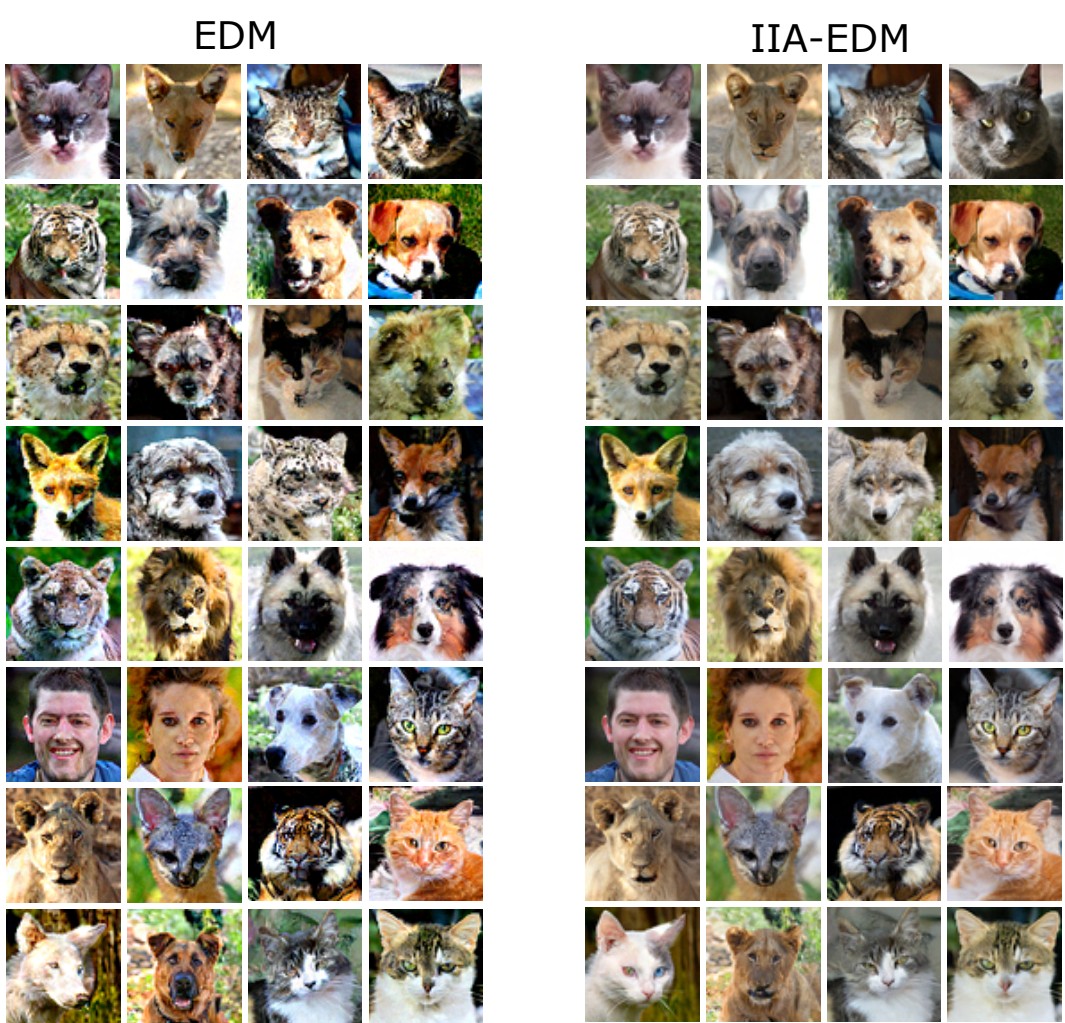

Figure 10: Comparison of images generated by EDM and IIA-EDM at 11 NFEs (or equivalently 6 timesteps).

