# OpenReview forum: "On Accelerating Diffusion-Based Sampling Processes via Improved Integration Approximation"
_ICLR.cc/2024/Conference — ICLR 2024 poster_

### Official Review · Reviewer_t2so · 2023-10-28

**Soundness:** 3 good
**Presentation:** 3 good
**Contribution:** 3 good
**Rating:** 6
**Confidence:** 3

**Summary:**

This paper proposes improved integration approximation (IIA) for diffusion model sampling. The core idea is to use past gradients in approximating the update term in diffusion ODEs. The parameters before past gradients are settled by minimizing MMSE to the fine-grained Euler approximation. The proposed IIA technique is only a sampling technique and does not require any modification of the pretrained model. Extensive experiments are conducted to illustrate the performance of this technique.

**Strengths:**

Originality: The authors studies a sampling technique in diffusion models. The key idea is to approximate ODE more precise to generate better images. This work focuses on a small task but the originality is good.

Quality: The presentation of this work is quite clear.

Significance: The technique is useful for generating high-quality images, and has potential to be integrated into SOTA diffusion realizations. However, as I'm not an expert in empirical diffusion models literature, the performance should be more carefully evaluated by other reviewers.

**Weaknesses:**

Some parts of the algorithm can be more clearly discussed:
 - In experiments (Table 2) the authors mostly use $r=1$, that is, only use past two gradients to estimate the integration term. It is better to discuss why to choose such a parameter, probably with some numerical illustrations. In addition, only for BIIA-EDM $r=0$ is chosen, I wonder if it is because of bad performance. Choice of fine-grained timesteps $M$ is also not discussed.
 - Personally I think the IIA idea resembles that of "Anderson Acceleration" in optimization literature, but that's not referred in the paper. The authors may gain some insights from it.
 - The authors could illustrate more about the difference between BIIA and IIA. It seems IIA only decomposes the gradient form $0.5d_i+0.5d'_{i+1|i}$ into two terms, and treat the coefficients of such two terms as optimization parameters so as to add flexibility. I wonder how the performance of IIA is compared to BIIA with larger $r$, i.e., with more previous approximations included in MMSE optimization.

**Questions:**

Discussed above.

---

> ### Author Response · Authors · 2023-11-16
>
> 1. In experiments (Table 2) the authors mostly use $r=1$, that is, only use past two gradients to estimate the integration term. It is better to discuss why to choose such a parameter, probably with some numerical illustrations.
>
> Thank you for this comment. In Appendix D.1 and D.2 of the revision, we explain through ablation studies that the setup $r=2$ does generally not improve the FID scores of the setup $r=1$ (see also our response to your Comment-5).
>
> 2. In addition, $r=0$ is chosen only for BIIA-EDM over FFHQ64, I wonder if it is because of bad performance.
>
> Thank you for the comment. For BIIA-EDM over FFHQ64, the setup $r=0$ performs slightly better than $r=1$. We emphasize that the performance for $r=1$ is still much better than that of EDM (see Table 3 in appendix D.1 in the revision for detailed FID comparison).
>
> 3. Choice of fine-grained timesteps $M$ is also not discussed.
>
> Thank you for the comment. Please refer to our response to Comment-4 of reviewer SzoU.
>
>
> 4. Personally I think the IIA idea resembles that of "Anderson Acceleration" in optimization literature, but that's not referred in the paper. The authors may gain some insights from it.
>
> Thank you for this insightful comment. We note that when we were preparing the manuscript, we were not aware of "Anderson Acceleration" in the optimization literature. We have briefly studied "Anderson Acceleration" in the last few days. The Anderson Acceleration considers minimizing a weight summation of a number of recent residual errors w.r.t. the weighting factors. The weighting factors are constrained to have a unit summation. In our work, we only consider minimizing one single residual error in the MSE for the current timestep w.r.t. to a set of coefficients. To obtain the smallest residual error, we avoid placing any constraint on the set of coefficients. Another difference is that in our work, the residual errors to be optimized are different for different timesteps. On the other hand, the list of recent residual errors being optimized by Anderson Acceleration essentially focuses on a single objective function.
>
> One future research direction would be to apply Anderson Acceleration to existing ODE solvers in diffusion process to see if it can perform better than IIA based ODE solvers.
>
>
> 5. The authors could illustrate more about the difference between BIIA and IIA. It seems IIA only decomposes the gradient form into two terms, and treats the coefficients of such two terms as optimization parameters so as to add flexibility. I wonder how the performance of IIA is compared to BIIA with larger $r$, i.e., with more previous approximations included in MMSE optimization.
>
> Thank you for the comment. We have performed an additional experiment to evaluate the performance of BIIA-EDM and IIA-EDM with $\gamma=2$ for CIFAR10. The results are summarized in Appendix D.2 of the revision. It is found that when $r$ increases from $r=1$ to $r=2$, the differences in the FID scores of each sampling method are very small. The performance gain of IIA-EDM over BIIA-EDM is still significant for $r=2$.

---

### Official Review · Reviewer_8gXX · 2023-10-30

**Soundness:** 3 good
**Presentation:** 3 good
**Contribution:** 3 good
**Rating:** 6
**Confidence:** 2

**Summary:**

The paper proposes an improvement to the diffusion model integration procedure in order to make the sampling procedure faster. In particular, in order to accelerate the numerical time integration over an interval t_i to t_{i+1}, an optimization problem is solved that optimizes the weights over gradients evaluated at a coarse grid, in order to minimize the error with  respect to the integration using a fine grid. Using the optimized weights, the inference procedure becomes faster by decreasing the number of steps, while the accuracy is kept consistent. The paper illustrates the performance of the algorithm, highlighting its computational gain, over several experiments.

**Strengths:**

- The paper discusses an important and timely topic.
- The contributions of the paper are clearly described
- Extensive numerical experiments are provided to support the contributions.

**Weaknesses:**

- The choice of several hyper-parameters is not clear. How does one choose the number and location of grid points for the coarse and find grids? What are the limitations? What will be the computational overhead if one has to search for suitable parameters? How does the computational gain vary as one changes these parameters?

- There is no discussion or comparison with other numerical integration methods. Numerical integration is a classical topic. It will be helpful to include discussion about why the existing methods are not appropriate.

**Questions:**

Please see the comments above.

---

> ### Author Response · Authors · 2023-11-16
>
> 1. The choice of several hyper-parameters is not clear. How does one choose the number and location of grid points for the coarse and find grids?
>
> Thank you for the comment. See our response to Comment-4 of reviewer SzoU.  In principle, a large value of $M$ would lead to a high accuracy of the integration approximation in the IIA based ODE solvers. On the other hand, a large $M$ value would require a high one-time computational overhead when performing MMSE. The additional ablation study in Appendix D.3 indicates that when $M$ is larger than a threshold (it is $M=10$ in Table~6 in the revision), the obtained FID scores and the averaged residual errors of the MSE remain roughly the same.
>
> In all of our experiments, the fine-grained timesteps were evenly distributed within each time-interval (see Appendix D and Section 5). We do not suggest to search for the optimal fine-grained timesteps, which is time-consuming. The resulting performance gain likely is marginal for the IIA based ODE solvers.
>
>
> 2. What will be the computational overhead if one has to search for suitable parameters? How does the computational gain vary as one changes these parameters?
>
> Thank you for the comment. Assume for a given $M$ value, the fine-grained timesteps within each coarse time-interval are selected to be evenly distributed instead of searching. In general, the one-time computational overhead would increase linearly as M increases for a pretrained model with a set of timesteps. This is because the NFEs increase linearly as $M$ increases (see the evidence of Table 6 in the revision). In practice, one can stop searching for $M$ when the residual error of the associated MSE remains roughly the same when $M$ increases. As an example, Table 6 indicates the time spent for investigating the performance for $M=20$ in a sampling procedure of 19 NFEs for a pre-trained model is roughly 3 minutes, which is acceptable from a practical point of view.
>
>
> 3. There is no discussion or comparison with other numerical integration methods. Numerical integration is a classical topic. It will be helpful to include discussion about why the existing methods are not appropriate.
>
> Thank you for the comment. We have added a new paragraph (in blue and red color) at the beginning of Section 3 in the revision. In the new paragraph, we argue that the format of existing state-of-the-art diffusion ODE solvers (e.g., EDM and DPM-Solver++) is fixed irrespective of the number of time-steps and pre-trained models. It is likely that the coefficients of ODE solvers are not always optimal. We then explain the basic design principle for incorporating IIA into existing ODE solvers.

---

> > ### Comment · Reviewer_8gXX · 2023-11-22
> >
> > I appreciate the author's response and answering my questions.

---

> > > ### Author Response · Authors · 2023-11-23
> > >
> > > Thank you for carefully checking our responses to your comments.

---

### Official Review · Reviewer_SzoU · 2023-10-31

**Soundness:** 3 good
**Presentation:** 4 excellent
**Contribution:** 3 good
**Rating:** 6
**Confidence:** 3

**Summary:**

This paper proposed to use the improved integration approximation(IIA) technique to numerically solve the reverse ODEs that appears in the ODE-based sampling processes, including EDM, DDIM and DPM-Solver. The authors introduced numerical algorithms based on IIA and explain the algorithms from both the theoretical and experimental perspective.

1. Theoretically, algorithms based on IIA are designed for EDM, DDIM and DPM-Solver. In these algorithms, gradient at each step is estimated by a linear combination of several most-recent gradients, which is expected to be more accurate than to only use one or two most recent gradients. The coefficients in the linear combination are obtained by solving an optimization problem, minimizing the MSE to some highly accurate integral approximation.

2. Experimentally, they verify the effectiveness of these algorithms on EDM, DDIM and DPM-Solver. It is observed that the IIA-based algorithms improves the sampling qualities for low NFEs($\le 25$).

**Strengths:**

1. This paper introduces new numerical algorithms for ODE-based sampling processes based on IIA. Although this paper focuses on EDM, DDIM and DPM-Solver, same idea applied to other ODE-based sampling processes as well.

2. The theoretical formulations of the algorithms are clear.

3. Experimental results are provided and they show the improvements of such algorithms in some cases(small NFEs).

**Weaknesses:**

1. The paper doesn't provide any theoretical result showing the effectiveness of the IIA-based algorithms.


2. According to the experiments, the improvement of the algorithms in FID only happen when NFE is small. It is also not clear why there is no significant improvement on FID with big NFE.

**Questions:**

Questions:

1. Comparing the integration approximation in $(9)$ and $(14)$, why do we preserve the factor $\frac{1}{2}$ in $(9)$ but ignore the two step-size in $(13)$ when we derive $(14)$? If we include the two $\frac{1}{2}$ factors into the coefficients in $(9)$, how would it affect the numerical results?

2. The MSE optimization is based on a high-accuracy integration approximation. How to choose the $M$ and fine-grained time-step in the high accuracy integration? Would the pre-trained time-step affect the parameters in the high-accuracy integration approximation?

Comment:

1. Typo in the second integral in $(10)$.

---

> ### Author Response · Authors · 2023-11-16
>
> 1. The paper doesn't provide any theoretical result showing the effectiveness of the IIA-based algorithms.
>
> Thank you for this comment. We agree that the paper does not provide a theoretical result quantifying to what extent the accuracy of integration approximation per timestep can be improved when applying the IIA technique to an existing ODE solver. On the other hand, we propose, for the first time in our paper, to compute the optimal stepsizes in an existing ODE solver by solving a set of quadratic optimization problems. In practice, one can easily verify the effectiveness of the IIA technique by checking if the residual errors of the quadratic functions after stepsize optimization are reduced notably or not without a need for computing the FID. Extensive results in our paper indicate that the sampling performance of DDIM, DPM-Solver++, and EDM can be improved significantly when NFE is small (e.g., less than 25). This implies that the coefficients in DDIM, DPM-Solver++, and EDM are not optimal at least for small NFEs even though they have strong mathematical motivations. The coefficients can advantageously be learned by data-driven methods such as the proposed IIA technique. We hope our work sheds light on opportunities for future research work on the design of more effective data-driven learning methods.
>
> 2. According to the experiments, the improvement of the algorithms in FID only happen when NFE is small. It is not clear why there is no significant improvement on FID with big NFE.
>
> Thank you for the comment. One major research challenge in diffusion models for the time being is how to improve the sampling quality when NFE is small. Such research work would lead to many industrial applications if the developed methods are effective. Our research work provides an efficient approach towards the above objective by solving a set of quadratic optmization problems for a pretrained model with a set of timesteps.   The performance gain of our approach is obtained by improving the accuracy of the integration approximation per timestep by means of stepsize optimization. In principle, when the NFE is large, the integration approximation of existing ODE solvers is relatively accurate, leaving little room to improve the accuracy further. Therefore, there is no significant improvement in FID with large NFE. We have added a new paragraph (in red color) at the end of Section 6 in the revision to explain this.
>
>
> 3. Regarding the factor $\frac{1}{2}$ in (9)
>
> Thank you for the comments. In principle, the factor $\frac{1}{2}$ can be absorbed into each coefficient $\gamma_{ik}$. We keep $\frac{1}{2}$ in the expression to make it more readable. Absorbing the two $\frac{1}{2}$ factors into each coefficient $\gamma_{ik}$ in (9) (which becomes (13) in the revision) will not affect the numerical results because after stepsize optimization, the residual error in the MSE should remain the same. The new optimal solution for each coefficient $\gamma_{ik}$ would simply be the original optimal solution scaled by $\frac{1}{2}$.
>
> 4. How to choose the $M$ and fine-grained time-step in the high accuracy integration? Would the pre-trained time-step affect the parameters in the high-accuracy integration approximation?
>
> Thank you for the comments. In principle,  a large value of $M$ would lead to a high accuracy of the integration approximation in the IIA based ODE solvers. On the other hand, a large $M$ value would also require a high one-time computational overhead when performing MMSE. In the revision, we performed an ablation study on how the $M$ parameter affects the performance of IIA-EDM for both 11 and 19 NFEs over CIFAR10. The results were summarized in Table 6 in the revised paper. It is found that when the $M$ value is above a threshold (it is $M=10$ in Table 6), the FID scores and the averaged residual errors remain roughly the same.  This implies that in practice there is  no need to make $M$ approach infinity. It is also found that the one-time computational overhead increases linearly with $M$.
>
> In all of our experiments, the fine-grained timesteps were selected to be evenly distributed within each time-interval for simplicity (see Appendix D and Section 5 for details). At the moment, we do not suggest searching for the optimal fine-grained timesteps within each time-interval. Firstly, one needs to come up with some criterion for the search procedure and it is not clear to us how to design such a criterion. Secondly, even if there exists a good criterion for the optimal fine-grained timesteps, we believe the performance gain is marginal for the IIA based ODE solvers.
>
>
> Finally, we do not fully understand your final question "Would the pre-trained time-step affect the parameters in the high-accuracy integration approximation?". It would be great if you could elaborate on this question.
>
>
> 5. Typo in the 2nd integral in (10)
>
> Thank you for the comment. We have corrected the typo in the revision.

---

> > ### Comment · Reviewer_SzoU · 2023-11-17
> >
> > Thanks for the response. I really appreciate the discussions added in the revision.
> >
> >
> > Regarding my question $4$,  I was asking if the fine-grained timesteps are not uniform, how would it affect the close-form solution to $\{\gamma^*_{ik}\}$?
> >
> > Overall, I agree with the authors that even if theoretical quantification of the improvement is not studied, the experimental improvement is significant and this paper can inspire future research along its direction. Based on this, I will willing to change my original rating to 6.

---

> > > ### Author Response · Authors · 2023-11-17
> > >
> > > Thank you very much for carefully reading our responses to your comments and for raising your score up.
> > >
> > > Regarding the performance for the case of non-uniform fine-grained timesteps, we think that if the number of fine-grained timesteps is sufficiently large and if the steps are properly distributed in each time-interval, the performance would also be good. The real question is what criterion should be used to optimize the non-uniform fine-grained timesteps. This is an interesting research question that we do not currently know the answer to.

---

### Official Review · Reviewer_an6f · 2023-10-31

**Soundness:** 3 good
**Presentation:** 3 good
**Contribution:** 2 fair
**Rating:** 6
**Confidence:** 4

**Summary:**

This work proposes IIA solvers by estimating the coefficients of diffusion ODE solvers. Instead of using the previous analytical coefficients, this work use the ground truth solution (by solving with many steps) at each interval to estimate the coefficients with a MSE loss, and then further accelerate the sampling procedure of diffusion ODEs, which can be understood as "distill the coefficients".

**Strengths:**

- The proposed method is easy to understand and the writing is easy to follow.
- The proposed method can be used for any previous ODE solvers and further improve them.

**Weaknesses:**

- Major:
  - The design for IIA solvers seem to be lack of a principle. For example:
    - In IIA-EDM, why $z_i-D_\theta$ is the "first gradient"? It is the gradient of what? Because the "2nd gradient" is the difference of two $D_\theta$, it is natural to understand the "first gradient" is $D_\theta$ itself, but not $z_i-D_\theta$. So what is the basic principle for designing it?
    - Why DDIM use data-pred model and noise-pred model for IIA? As data-pred model can be equivalently rewritten to noise-pred model, it seems to be equivelent to a linear combination of $z_i$ and $\epsilon_\theta(z_i, t_i)$.
- Minor:
  - Table 1, Column 6, should be "IIA-DPM-Solver" instead of "IIA-PDM-Solver".
  - As far as I known, the dpm variant for SD v2 is 2nd-order multi-step DPM-Solver++, not DPM-Solver. Please clarify the detailed setting.

**Questions:**

Please clarify the common design principle for IIA for different solvers. Is there any common principle such that we do not need to one-by-one design them?

====================

Thanks for the authors' revisions. I think the revised version addressed my concerns to some extent, so I raise my score to 6.

---

> ### Author Response · Authors · 2023-11-16
>
> 1. The design for IIA solvers seem to be lack of a principle?
>
> Thank you for the comment. We have refined our paper accordingly. In the revision, we have added a new paragraph at the beginning of Section 3 to explain the basic design principles for incorporating IIA into existing ODE solvers. Basically, we point out that the selected quantities in an IIA-based ODE solver are appropriate as long as the residual error of the MSE after stepsize optimization is reduced notably compared to that of the original ODE solver. When we incorporate IIA into DDIM, DPM-Solver++, and EDM by following the above guidance, we focus on a particular proper selection of quantities for each sampling method instead of finding the
> set of quantities that produces the smallest residual error. The particular design of IIA-DDIM, IIA-DPM-Solver++, and IIA-EDM in our paper is because the resulting residual errors are reduced notably compared to those of the original ODE solvers. We leave the research for finding the optimal set of quantities to future work.
>
>
> 2. In IIA-EDM, why $z_i-D_{\theta}$ is the "first gradient"? It is the gradient of what? Because the "2nd gradient" is the difference of two $D_{\theta}$, it is natural to understand the "first gradient" $D_{\theta}$ is itself, but not $z_i-D_{\theta}$.
>
> Thank you for this comment. In the revision for IIA-EDM (see (17) in the revised paper), we have changed "1st gradient" to "1st quantity" and "2nd gradient" to "2nd quantity" to avoid any ambiguity.  It is noted that the original stepsizes in front of the two quantities are $\frac{t_{i+1}-t_i}{t_i}$ and $\frac{t_{i+1}-t_i}{2t_{i+1}}$ at timestep $t_i$. We argue that we can optimize the two stepsizes in front of the two quantities by MMSE. As we explained earlier, the basic design principle is to ensure that after stepsize optimization, the resulting residual error of the corresponding MSE is reduced. It is clear from Fig. 4 in the paper that the average residual error after stepsize optimization in IIA-EDM is reduced significantly compared to that of EDM.
>
> 3. Why DDIM use data-pred model and noise-pred model for IIA? As data-pred model can be equivalently rewritten to noise-pred model, it seems to be equivalent to a linear combination of $z_i$ and $\theta_{\theta}(z_i, t_i)$.
>
> Thank you for the comments.  We have conducted additional experiments in the revision showing that IIA-DDIM with data-pred models alone also works well in practice. The results are summarized in Appendix D.4. In particular, we investigated the FID scores and the averaged residual errors of IIA-DDIM with one quantity and two quantities. The IIA-DDIM with one quantity only considers the difference $\hat{x}(z_i, t_i)-\hat{x}(z_{i-1}, t_{i-1})$ of data-pred models at timestep $t_i$ while IIA-DDIM with two quantities considers both  $\hat{x}(z_i, t_i)-\hat{x}(z_{i-1}, t_{i-1})$ and $\epsilon_{\theta}(z_i, t_i)-\epsilon_{\theta}(z_{i-1}, t_{i-1})$ at timestep $t_i$. It is found that IIA-DDIM with two quantities performs slightly better than IIA-DDIM with one quantity w.r.t. both the FID scores and the averaged residual errors. We also show in Appendix D.4 (see (26)) that $\hat{x}(z_i, t_i)-\hat{x}(z_{i-1}, t_{i-1})$ cannot be represented in terms of $\epsilon_{\theta}(z_i, t_i)-\epsilon_{\theta}(z_{i-1}, t_{i-1})$ and $(z_i, z_{i-1})$.
>
> 4. Table 1, Column 6, should be "IIA-DPM-Solver" instead of "IIA-PDM-Solver".
>
> Thank you for pointing out the typo. We have corrected it in the revision to be "IIA-DPM-Solver++".
>
> 5. As far as I known, the dpm variant for SD v2 is 2nd-order multi-step DPM-Solver++, not DPM-Solver. Please clarify the detailed setting.
>
> Thank you for spotting the typos in our paper. It should be multi-step DPM-Solver++ instead of multi-step DPM-Solver. We have corrected the typos in the main paper and Appendix C which describes the design principle of IIA-DPM-Solver++.

---

> > ### Comment · Reviewer_an6f · 2023-11-22
> > **Thank you for the rebuttal!**
> >
> > Thanks for the authors' revisions. I think the revised version addressed my concerns to some extent, so I raised my score to 6.

---

> > > ### Author Response · Authors · 2023-11-22
> > >
> > > Thank you very much for carefully checking the modifications introduced in the revised version and for raising your score up.

---

### Author Response · Authors · 2023-11-16
**Summary of major changes in the revised paper**

Dear reviewers,

Firstly, thank you for finding that our proposed IIA technique is innovative and promising for improving the performance of existing ODE solvers in diffusion models for small NFEs (e.g., less than 25).

(1): We have added a new paragraph (in blue and red color) in the beginning of Section 3 of the revised paper, studying the limitations of existing ODE solvers and explaining the design principle of the IIA technique. Basically, the selected quantities are appropriate when applying IIA to an existing ODE solver as long as the residual error of the MSE after stepsize optimization is reduced notably compared to that of the original ODE solver.  (2): We have added a new paragraph (in red) at the end of Section 5, explaining why there is no large FID improvement when applying the IIA technique to an existing ODE solver for large NFEs. (3): We have performed a number of additional ablation studies of the IIA technique in Appendix D. In particular, we have studied how the $r$ parameter affects the performance of BIIA-EDM and IIA-EDM by using a pre-trained model for CIFAR10. We have also examined how the $M$ parameter affects the performance of IIA-EDM by using a pre-trained model for CIFAR10. It is found from the ablation study that once $M$ is larger than a threshold, the obtained FID scores and averaged squared residual errors remain constant. Finally, we have investigated the performance of IIA-DDIM with one quantity (i.e., using only estimated clean data) and two quantities (i.e., using both the estimated clean data and estimated Gaussian noises) using a pre-trained model for CIFAR10.

---

### Meta-Review · Area_Chair_g8Q1 · 2023-12-05

**Metareview:**

This paper focuses on the numerical simulation of probabilistic flow, which is the ODE version of the backward process of pre-trained diffusion generative model. A numerical integration method is proposed for accelerated generation speed, based on optimizing certain coefficients of the numerical integration scheme at each time step. Reviewers and AC reached a consensus that this idea is interesting despite of the existence of minor concerns. Therefore, acceptance is recommended. However, AC urges the authors to revise the paper where appropriate, based on the discussion with the reviewers.

**Justification For Why Not Higher Score:**

Multiple reviewers raised concerns such as missing a theoretical comparison of the proposed method against existing numerical schemes, and I agree.

**Justification For Why Not Lower Score:**

Reviewers unanimously recommended acceptance and I agree with their rationale.

---

### Decision · Program_Chairs · 2024-01-16

Accept (poster)